# Overcoming Treatment Challenges in Posterior Segment Diseases with Biodegradable Nano-Based Drug Delivery Systems

**DOI:** 10.3390/pharmaceutics15041094

**Published:** 2023-03-29

**Authors:** Kevin Y. Wu, Maxine Joly-Chevrier, Dania Akbar, Simon D. Tran

**Affiliations:** 1Department of Surgery, Division of Ophthalmology, University of Sherbrooke, Sherbrooke, QC J1G 2E8, Canada; yang.wu@usherbrooke.ca; 2Faculty of Medicine, University of Montreal, Montreal, QC H3T 1J4, Canada; 3Department of Human Biology, University of Toronto, Toronto, ON M5S 1A1, Canada; 4Faculty of Dental Medicine and Oral Health Sciences, McGill University, Montreal, QC H3A 1G1, Canada

**Keywords:** ocular surface disease, retinal disease, nanosystems for ocular drug delivery, nanocarriers, biodegradable polymers, ocular drug delivery system, hydrogels, ocular inserts, exosomes

## Abstract

Posterior segment eye diseases present a challenge in treatment due to the complex structures in the eye that serve as robust static and dynamic barriers, limiting the penetration, residence time, and bioavailability of topical and intraocular medications. This hinders effective treatment and requires frequent dosing, such as the regular use of eye drops or visits to the ophthalmologist for intravitreal injections, to manage the disease. Moreover, the drugs must be biodegradable to minimize toxicity and adverse reactions, as well as small enough to not affect the visual axis. The development of biodegradable nano-based drug delivery systems (DDSs) can be the solution to these challenges. First, they can stay in ocular tissues for longer periods of time, reducing the frequency of drug administration. Second, they can pass through ocular barriers, offering higher bioavailability to targeted tissues that are otherwise inaccessible. Third, they can be made up of polymers that are biodegradable and nanosized. Hence, therapeutic innovations in biodegradable nanosized DDS have been widely explored for ophthalmic drug delivery applications. In this review, we will present a concise overview of DDSs utilized in the treatment of ocular diseases. We will then examine the current therapeutic challenges faced in the management of posterior segment diseases and explore how various types of biodegradable nanocarriers can enhance our therapeutic arsenal. A literature review of the pre-clinical and clinical studies published between 2017 and 2023 was conducted. Through the advances in biodegradable materials, combined with a better understanding of ocular pharmacology, the nano-based DDSs have rapidly evolved, showing great promise to overcome challenges currently encountered by clinicians.

## 1. Introduction

The treatment of posterior segment eye diseases presents a major challenge due to the complex anatomy of the eye, which acts as a barrier to the effective delivery of medications. Conventional treatments, such as topical eye drops or intravitreal injections, are limited by their poor bioavailability and short residence time, requiring frequent dosing to manage the disease. To address these limitations, researchers have turned to the development of biodegradable nano-based drug delivery systems (DDSs) as a potential solution. These systems offer longer residence time in ocular tissues and better penetration through ocular barriers, and they are made up of biodegradable polymers of nanosized dimensions, reducing the risk of toxicity and adverse reactions.

In this review article, we provide a comprehensive overview of the latest advances in biodegradable nano-based DDSs for the treatment of posterior segment diseases (Figure 1). By examining the current therapeutic challenges and exploring the various types of biodegradable nanocarriers, we aim to highlight the potential of these systems to enhance the treatment of posterior segment diseases. Through a literature review of pre-clinical and clinical studies published between 2017 and 2022, we demonstrate the rapid evolution of nano-based DDSs. With the rapid evolution of biodegradable materials and a deeper understanding of ocular pharmacology, nano-based DDSs have shown great promise in overcoming the obstacles currently faced by ophthalmologists.

## 2. Anatomical Barriers in Ocular Drug Delivery

There are several methods for administering ophthalmic medications to ocular tissues, including, but not limited to, topical, subconjunctival, periocular, intravitreal, and systemic delivery [1]. Topical, systemic, and intraocular delivery are the three main routes to deliver drugs to the back of the eye. The simplest method remains topical application in the form of various preparations, such as solutions, suspensions, ointments, gels, or emulsions [2]. However, only a small portion of the applied dose, about 5%, can penetrate the internal structures of the eye. The uptake of solutes and fluids into the anterior and posterior parts of the eye is restricted by ocular barriers, such as the tear film, cornea, conjunctiva, vitreous, blood-aqueous barrier, and blood-retina barrier (Figure 2). These barriers protect the eye from potentially harmful molecules from the external environment, but at the same time, they also reduce the bioavailability of ocular drugs [2]. Notably, the blood-retina barrier is largely responsible for limiting drug absorption to the posterior parts of the eye, while the other barriers are primarily responsible for opposing absorption into the anterior parts of the eye [3].

### 2.1. Tear Film in Ocular Drug Delivery

The tear film is characterized by a complex structure of three layers: a lipid layer, an aqueous layer, and a mucin layer, which all rest atop the hydrophobic surface of the epithelium. Microscopically, there is no clear demarcation between the mucous and aqueous layers [4].

One major challenge in topical eyedrop delivery is the constant removal of the drug from the eye surface by the lacrimal fluid secretion. This clearance mechanism, along with reflex blinking, contribute to significant tear turnover. The lacrimal fluid turnover rate of approximately 1µL/min can result in a rapid removal of significant drug doses [2].

### 2.2. Nasolacrimal Drainage System in Ocular Drug Delivery

A significant portion of a drug instilled into the eye, approximately 95%, is eliminated from the ocular surface and cul-de-sac through the nasolacrimal duct. This duct acts as a pathway for tear outflow from the eye to the nasal cavity. The nasolacrimal drainage system includes the lacrimal sac, canaliculi, and nasolacrimal ducts. The issue is that the vascularized walls of the lacrimal sac and nasolacrimal duct contribute significantly to the undesirable systemic absorption of the drug, leading to both systemic side effects and reduced availability of the topical drug for the targeted ocular tissue [5]. The undesirable systemic absorption rate provided by the nasolacrimal drainage system is influenced by factors such as the volume of the topical drug solution, the patient’s reflex blinking, and age. To address this challenge, the design of the drug delivery should prioritize retention on the ocular surface to efficiently deliver topical medications to targeted ocular structures while minimizing the amount drained into the nasolacrimal drainage system [2].

### 2.3. Cornea in Ocular Drug Delivery

The cornea, composed of five layers—the epithelium, Bowman’s membrane, stroma, Descemet’s membrane, and endothelium—serves as a mechanical and chemical barrier, limiting the access of exogenous substances into the eye and protecting intraocular tissues [6]. Tight junction complexes are present in the superficial epithelial cells, while gap junctions are found in the wing and basal cells. The stroma and Descemet’s membrane cover the inner endothelial cells, which contain macula adherents while simultaneously allowing the transverse of materials [2].

The cornea is a semi-permeable membrane that passively allows material transfer across its cells. The tight junctions (zonulae occludens) on the surface of the corneal epithelium prevent the diffusion of macromolecular and hydrophilic molecules, allowing only relatively small molecules to permeate through the pores (average diameter of 2.0 nm). The stroma, with its high percentage of hydrated collagen, is more hydrophilic and hinders the transverse wave of lipophilic molecules. The negatively charged pores at physiological pH also pose a challenge for charged molecules to ionic interaction. The transcorneal transfer is influenced by factors such as lipophilicity, molecular weight, charge, and the degree of ionization of the drug. In some cases, despite successful diffusion into the aqueous humor after transcorneal transport, drugs are unable to reach the posterior portions of the eye at therapeutic concentrations due to reduced diffusion across the vitreous humor [3].

In contrast, intravitreal drug administration provides a more direct path to the vitreous and retina. However, the diffusion of larger and positively charged drugs across the RPE barrier to the choroid may be impeded [3].

Drug elimination from the aqueous humor happens through two pathways: aqueous turnover through the chamber angle and Schlemm’s canal (also known as the conventional trabecular meshwork outflow pathway), and venous blood flow from the anterior uvea (also known as the uveoscleral outflow pathway) [7]. The former method relies on convective flow and is independent of the drug’s properties, while the latter is dependent on the drug’s lipophilicity as it must cross the endothelium of the vessels before being eliminated by uveal blood flow [1].

### 2.4. Blood–Ocular Barrier (BOB)

The blood–ocular barrier (BOB) is composed of the blood–aqueous barrier (BAB) and the blood–retinal barrier (BRB), both of which serve as major impediments to systemic and topical drug delivery in the eye’s anterior and posterior chambers [8].

#### 2.4.1. Blood–Aqueous Barrier (BAB)

The BAB, related to the anterior chamber, consists of endothelial cells, iris, ciliary muscle, and pigmented and nonpigmented epithelium cells, and is characterized by tight junctions that restrict drug molecule entry [2].

#### 2.4.2. Blood–Retinal Barrier (BRB)

The BRB further hinders drug entry from the blood into the posterior chamber, and comprises retinal capillaries and the retinal pigment epithelium cells (RPEs) as the inner and outer blood–retinal barriers, respectively [9]. Drug permeability across RPEs is easier to determine, but the permeability of retinal capillaries is harder to quantify. Additionally, particle size is a crucial factor in the permeation of drugs through retinal capillaries [2].

## 3. Routes of Administration for Treating Ocular Diseases

Topical administration is the most commonly used non-invasive route for ocular drug delivery, but it has low bioavailability due to short residence time and poor corneal permeability. Topical drugs require frequent dosing with high concentrations, which can lead to ocular and systemic side effects and affect patient compliance. Moreover, this method can be difficult for handicapped and elderly patients [10,11,12].

Intravitreal injections have gained attention for delivering ocular drugs due to their ability to offer superior bioavailability to the posterior segment of the eye. Recent studies have shown the effectiveness of intravitreal drug delivery systems such as rapamycin-loaded polymeric micelles and nano-liposomes of vancomycin [13,14]. Other intravitreal drug delivery methods include the use of vitamin E/poly-lactic-co-glycolic acid microspheres containing glial cell line derived neurotrophic factor (GDNF) to protect retinal ganglion cells and biodegradable polyester amide implants for diabetic macular edema and neovascular age-related macular degeneration treatment [15,16,17]. An intravitreal injection of flurbiprofen using a novel liposome aggregate platform demonstrated prolonged drug retention and reduced inflammation in ocular tissue compared to conventional liposomes [18]. These studies demonstrate the potential of intravitreal drug delivery for sustained drug release with predictable pharmacokinetic profiles.

For some posterior segment diseases that cannot be treated by conventional eye drops, intravitreal injections, and systemic drug delivery, posterior juxtascleral injections can be an alternative. Anecortave cortisone administered through juxtascleral injection provided a sustained release up to 6 months for age-related macular degeneration treatment [19]. The juxtascleral injection of hollow microcapsules loaded with an anti-vascular endothelial growth factor protein for macular degeneration and diabetic retinopathy treatment can form a depot on the episcleral surface [20]. Additionally, transscleral microneedles have been designed for retinal gene therapy [21].

Periocular injections include subconjunctival, sub-Tenon, and retrobulbar injections. Periocular injections can bypass the corneal and conjunctival barriers, allowing drugs to reach therapeutic levels behind the lens–iris diaphragm. Periocular injections are useful for drugs with low lipid solubility as they do not penetrate the eye adequately if they are given topically.

Subconjunctival injection is a preferred route of drug delivery when topical drops cannot penetrate the anterior segment of the eye. Low particle size brinzolamide-loaded PLGA nanoparticles, dorzolamide-loaded polyether anhydride microparticles, and thermogel with voriconazole have shown sustained drug release and significant efficacy [22,23,24,25]. Triamcinolone acetonide injection was more effective in preventing inflammation after cataract surgery than steroidal eye drops, while subconjunctival injection of human mesenchymal stromal cells reduced corneal inflammation in mice with graft versus host disease [26,27].

In addition to reaching therapeutic levels behind the lens–iris diaphragm, sub-Tenon injections can also deliver medication closer to the local site of action. For instance, posterior sub-Tenon injections of steroids can be used to treat cystoid macular edema (CME).

Although retrobulbar injection is mostly used for anesthesia purposes (i.e., retrobulbar block), this injection is a route of drug delivery that can enter the globe similarly to subconjunctival injection. For example, a single retrobulbar injection of Amphotericin B has been shown to effectively treat rhino-orbital cerebral mucormycosis with associated anterior cerebritis, whereas intravenous treatment was not as effective [28,29].

Intracameral injections can be used after cataract surgery and can be injected directly into the anterior chamber [30]. Iontophoresis uses a voltage gradient to deliver drugs to the back of the eye and has been shown to be effective for delivering nanoparticles and drugs loaded in contact lenses [31,32]. Iontophoretic delivery has also been developed for pilocarpine and besifloxacin-loaded liposomes [33,34].

The suprachoroidal space (SCS) is being explored as a potential approach to target pharmacotherapies to the posterior segment via a minimally invasive injection procedure. Clinical trials have explored the efficacy and safety of suprachoroidal injection of pharmacologic therapies in conditions affecting the posterior segment, with promising results for non-infectious uveitis. Suprachoroidal administration also shows potential for other applications, such as an injection of antiglaucoma agents into the anterior SCS and delivery of gene- or cell-based therapies for retinal disorders [35].

Figure 3 illustrates the different routes of administration for ophthalmic medications, including topical, subconjunctival, suprachoroidal, intracameral, intravitreal, retrobulbar, sub-tenon, posterior juxta scleral, subretinal, and systemic administration.

## 4. Overview of Biodegradable Nano-Based Drug Delivery System (DDS)

### 4.1. Biodegradable Nanocarriers for Improved Drug Delivery in Ocular Formulations

Conventional ocular drug formulations face challenges such as low bioavailability and quick clearance, leading to the need for frequent high-dose administrations, which can result in reduced patient compliance and increased side effects. Biodegradable nano-based DDSs offer several benefits to address these limitations. For example, the use of polymers such as viscosity enhancers can improve drug bioavailability by increasing retention time in the tear film without causing blurred vision [2]. Mucoadhesive polymers can also reduce lacrimal clearance by electrostatically binding to negatively charged mucin. Furthermore, nanocarrier DDSs can penetrate ocular barriers to reach the target site, with the aid of permeation enhancers. Targeting moieties can be added to prevent non-specific drug distribution, and targeted delivery strategies can respond to disease-specific stimuli to release drugs only in target tissue [36]. Lastly, nano-based DDSs can enhance drug stability in the eye by reducing interactions with tear proteins and avoiding electrostatic modifications from the different pH levels in eye structures [37].

### 4.2. Ideal Properties of Nanocarriers

Nanocarriers are a promising tool for drug delivery as they can enhance the bioavailability and efficacy of drugs while minimizing potential side effects. To be effective, nanocarriers should possess certain characteristics. Figure 4 outlines the ideal characteristics of nanocarriers for drug delivery.

Firstly, nanocarriers should be biodegradable and biocompatible, meaning they can be broken down by the body and do not cause any harm to cells or tissues. Additionally, they should have a uniform size distribution to ensure consistent drug delivery.

Furthermore, nanocarriers should be stable in biological environments to avoid premature drug release and ensure the drug reaches its intended target. They should also have controlled cellular uptake to optimize drug delivery and improve the pharmacodynamic effect of the drug.

Other important characteristics of nanocarriers include feasibility for large-scale production, high payload capacity, and controlled release. Ease of upscaling to bulk manufacturing and cost-effectiveness are also important considerations in the development of nanocarriers for drug delivery. By meeting these ideal characteristics, nanocarriers have the potential to significantly improve drug delivery and patient outcomes.

Blending multiple polymers into nanoassemblies is a strategy to achieve optimal physiochemical properties, like surface charge, solubility, and aggregation. Copolymers offer greater benefits than homopolymers, proteins, or lipids due to their tunable properties, low toxicity, and ability to functionalize. Supramolecular assemblies, cross-linked hydrogels, and block copolymerization are common methods for nanocarrier synthesis, with the morphological properties (sphere, rod, cylinder, etc.) affecting the final properties [37].

### 4.3. The Importance of Biodegradability and Morphology in Nano-Based DDS

To achieve effective and safe drug delivery, the polymers used in nanocarriers must possess the key properties mentioned in the last section, including biodegradability. Biodegradability is a crucial attribute as it ensures that the nanocarriers will degrade into biocompatible metabolites, thereby avoiding the potential risks and complications that can occur with the surgical removal of non-biodegradable delivery systems. In other words, biodegradability is a critical factor in the design of nanocarrier drug delivery systems, as it ensures the safe and effective delivery of drugs to the target site [38].

The morphology of the nanocarriers also plays a crucial role in drug release, with biodegradability being an important factor in determining the optimal morphological design. There are two primary mechanisms of drug release from nanocarriers: hydrophilic porous release and surface erosion. Hydrophilic porous release occurs when the nanocarrier has hydrophilic pores that allow for water diffusion and displacement of the drug. In contrast, surface erosion involves the use of specific polymers that allow for the gradual erosion of the nanocarrier surface, thereby releasing the drug in a controlled manner. This mechanism is especially important for the protection of water-sensitive drugs, as it minimizes exposure to water and maximizes stability [36].

### 4.4. The Different Biodegradable Polymers and Their Advantage

The use of biocompatible and biodegradable polymers in ocular drug delivery is becoming increasingly popular. Some of the common biodegradable polymers used in this field include hyaluronic acid, cellulose, chitosan, alginate, poly(lactide-co-glycolide) (PLGA), poloxamers, and cyclodextrins. Hyaluronic acid can retain water, which makes it ideal for hydrogel formulations [39]. Cellulose is commonly used to improve the viscosity of eye drops and has the advantage of ease in bulk manufacturing [40,41,42,43]. Chitosan is mucoadhesive and has in situ gelling properties, making it suitable for ocular drug delivery applications. Alginate is commonly used in ocular drug delivery as a copolymeric nanoparticle due to its ability to interact with eye membranes [44,45]. PLGA is widely used in drug delivery because of its ability for modifications that alter its size and surface potential [46,47]. Poloxamers are biodegradable surfactants composed of polyethylene and polypropylene oxide blocks and are used as a vehicle for ocular drug delivery. Cyclodextrins are cyclic oligosaccharides with hydrophobic cavities and hydrophilic surfaces that enhance the bioavailability of drugs and have potential for ocular drug delivery when functionalized [3,7,8].

### 4.5. Types of Nano-Based Drug Delivery Systems: Characteristics and Enhancements

Nano-based DDSs have been developed to promote sustained and selective drug delivery to different anatomical targets in the eye. By modifying the interactions between the different biodegradable polymers, the drug delivery of nanocarriers could be tailored to the specific physicochemical properties of the guest molecules and the microenvironment of the biological target. Figure 5 gives a summary of the characteristics.

#### 4.5.1. Nanomicelles

Nanomicelles are spherical structures made up of surfactant molecules. They spontaneously self-assemble in water or other polar solvents. The hydrophobic tails of the surfactant molecules point inward towards the center of the micelle, while the hydrophilic heads point outward towards the solvent. This structure makes nanomicelles ideal for encapsulating hydrophobic compounds and delivering them to specific locations in the body. Increasing the concentration of amphiphilic polymers results in the formation of nanomicelles with higher encapsulation capacity. Polymeric micelles have a lower critical micelle concentration and maintain a more stable shape. Recently, the addition of mucin-targeting moieties (such as cyclic peptide ligand, phenylboronic acid etc.) has been shown to enhance ocular micro-adhesion [48]. Moreover, the further shrinking of nanomicelles [49] has been reported to increase their efficiency. An important barrier to the clinical translation of nanomicelles is their premature degradation in systemic administration, [50] which is primarily circumvented with topical ocular administration. Cross-linking approaches for polymeric nanomicelles can enhance the stability of micelles to avoid premature drug release [51] and can be applied for stimulus-responsive release in topical ocular administration [52].

In addition to enhancing corneal penetration, nanomicelles have been designed to deliver drugs to the posterior segments of the eye. In vivo studies of Chitosan oligosaccharide–valylvaline–stearic acid nanomicelles revealed their high ability to reach the posterior segments through the conjunctival route [53].

#### 4.5.2. Liposomes

Liposomes are a similar formulation that have been most used in ocular drug deliveries. They are vesicles composed of one or more phospholipid bilayers that have advantageous pre-corneal and conjunctival penetration [34,54]. Importantly, the liposomal constituents are very flexible for chemical modification, which can be tailored to the physiochemical properties of the guest and the biological microenvironment. Altering the head group charge using positively charged stearyl amine or negatively charged diacetyl phosphate has been used in ocular drug delivery to selectively change their interactions with mucin and corneal permeability [55]. An important method to modify the surface of liposomes to make them more stable and better suited for certain applications is the use of polyamidoamine (PAMAM) coats. PAMAM is a type of synthetic polymer that is biocompatible and commonly used. PAMAM-coated compound liposomes can penetrate the corneal barrier and move relatively quickly in the corneal epithelium and have also been shown to raise the bioactivity of the guest compound by 1.6 times compared to normal liposomal formulations [54]. Liposomes are known as easily biodegradable molecules in the human body. Their biodegradability mechanisms consist of phagocytosis by the mononuclear phagocyte system cells. Opsonin proteins contained in the serum attach themselves to the liposomes and activate the complement system, which leads to an uptake by the macrophages [56].

#### 4.5.3. Dispersed Nanoparticles

Another important strategy to solubilize different host molecules is using nanoparticles. Supramolecular assemblies are formed by the combination of two or more molecular subunits governed primarily by van der Waals force, hydrophobic interactions, electrostatic interaction, hydrogen bonding, and cation–π interaction. The self-assembly and drug release of many nanoparticles are governed by these molecular interactions, and their development has gained extensive attention over the past few years [57]. The reduced size and lack of aggregation of dispersed nanoparticles increase their carrying efficiency by allowing for an increase in surface binding sites within the same molar amount of nanoparticle. Furthermore, their ability to selectively accumulate in tissue through the enhanced permeability, retention effect, and mucoadhesive characteristic has made them very attractive for commercial ocular applications.

Nanoparticles form self-assembling and ordered architectures with different topologies tailored to encapsulate the guest drugs including nanospheres, nanorod assemblies [15,16], and nanocapsules. These polymers usually respond rapidly to micro-environment changes and can release guest molecules in controlled schedules, which makes them good platforms for drug-delivery carriers. Nanorod formulations have more desirable drug delivery properties for ocular applications compared to nanospheres due to a lower uptake by macrophages and a greater half-life [58,59].

Nanocapsules are made up of a core of the drug surrounded by a protective coating, typically made of polymers. The small size of the nanocapsules allows them to easily penetrate the eye’s barriers and reach the target tissue, while the protective coating prevents rapid degradation and increases residence time in the eye. Nanocapsules have been used to deliver a variety of drugs, including anti-inflammatory and anti-glaucoma agents [60].

Solid Lipid Nanoparticles (SLNPs) are emerging constructs that utilize a solid lipid as the core material, surrounded by a stabilizing surfactant. This solid lipid core gives them many advantages over traditional liposomes and microemulsions, including better stability, improved drug loading capacity, and ease of manufacturing. Additionally, SLNPs have higher thermal stability and longer shelf life compared to most nanoparticles [61].

#### 4.5.4. Dendrimers

Dendrimers are repeating multibranched polymers that have a characteristic core with high density and precise functional groups attached to their surface. Compared to the different linear polymers, dendrimers have a very high encapsulation efficiency and a more predictable biodistribution profile. Due to their core-shell architectures, dendrimers maintain narrow polydispersity, which allows them to have promising potential for ocular DDS applications [62]. The multifunctional cores and well-defined nanostructure make dendrimers an ideal building block for synthesizing three-dimensional cross-linked networks called dendrimer hydrogels. The degree of cross-linking and gel properties can be adjusted readily by controlling reactant concentration or functional group density on the dendrimer surface. Dendrimers can serve as the building blocks for nanogels or liposomes, both of which have been shown to have large internal hydrophobic pores with superior loading capacity for ocular drug delivery [54,61].

#### 4.5.5. Hydrogels

Hydrogels have recently been investigated extensively for application in ocular drug delivery. Polyacrylic acid is known to exhibit a strong drug release at a pH of 5.5 and can be combined with different pH-responsive polymers to remain mechanically strong and achieve long-term drug release. Ideally, a drug delivery system should release the drug in response to the physiological need of the body. Thus, various chemical modifications (for example, amine, ortho ester, imine, and hydrazone) have been used to alter the pH sensitivity of different hydrogels. Furthermore, cellulose, chitosan, N-isopropylacrylamide, Poloxamers, PLGAs, and polyethylene glycols (PEGs) have been used to construct thermo-responsive DDSs with the advantage of being injectable at room temperature and rapid transition to gels at higher temperatures [63]. Ultrasound-responsive hydrogels can also prevent the downside of off-target release. The application of ultrasound waves has proven to be beneficial in the penetration of drugs through the anatomical barriers of the eye [64].

The extended drug release profiles of hydrogels can delay the frequency of intravitreal injections. Moreover, the addition of designated targeting moieties and drugs into the hydrogel in the sol state can allow the targeted delivery of the drugs to the site of action. Thus, hydrogels can provide a system that can deliver drugs to the posterior segments of the eye while lowering the risks of frequent ejections.

#### 4.5.6. Nanosuspensions and Nanoemulsions

Nanosuspension and nanoemulsions have emerged as approved DDSs for ocular delivery applications [65] due to their unique dispersion of poorly soluble and permeable drugs. Currently, nanoemulsions remain among the most approved nano-based biodegradable ocular DDS. They are very small droplets of one liquid dispersed within another liquid. One of the main advantages of using nanoemulsions for ocular drug delivery is that they can improve the solubility and stability of poorly soluble drugs, making them more suitable for ocular administration. The use of amphiphilic salts of cholesterol can greatly stabilize these formulations, making them effective for the delivery of antiglaucoma, anti-inflammatory, and antiviral drugs [66]. Increasing the fraction of the dispersed oil phase can increase the viscosity of the nanoemulsions, making them more bioavailable at their target site. Furthermore, the addition of water-soluble polymers can allow for the formation of a gel with a high retention time.

Nanosuspensions are aqueous dispersions of insoluble drug particles stabilized by surfactants. They are especially important when a drug molecule has a large molecular weight and dose, high melting point, and inability to form salts [66]. Nanosuspensions are advantageous due to their ability to circumvent high osmolarity produced by ophthalmic solutions while prolonging drug release profiles. Nanosuspensions are frequently prepared in an aqueous medium and, thus, their chemical stability can be affected by hydrolysis. Using other polymers to stabilize the nanosuspensions can prevent their aggregation and provide steric repulsion to make the structures physically stable. The stabilizers that have been investigated include poloxamers, phospholipids, and different cellulose derivatives [67].

#### 4.5.7. Microneedles

Finally, microneedles have been used in ocular drug delivery to facilitate the deposition of drugs in the subarachnoid and posterior eye segments. Compared to subconjunctival injections, microneedles can be self-administered, deliver the drug guest more accurately to the target site, and avoid the complications observed when using injections that may compromise the integrity of the eye barriers. Hollow microneedles have been synthesized to load greater concentrations of guest molecules, which can be further altered by changing the grafting ratios of the copolymers [68].

## 5. Posterior Segment Diseases

For delivering treatments to the posterior segment of the eye, bioavailability becomes a challenge using a systemic or topical route. Most systemic medications, whether administered orally or intravenously, have difficulty crossing the blood–retinal barrier (BRB), requiring a high-dose administration that can result in systemic side effects. Topical eye drops also have limitations due to the multiple ocular barriers that impede the medication’s path from the ocular surface to the posterior segment, as discussed in a previous section of this article. Intravitreal injections are more invasive and can lead to potential sight-threatening complications such as endophthalmitis or retinal detachment. Additionally, they have a short retention time and require multiple visits to the ophthalmologist for administration in a sterile condition, which decreases patient compliance [69]. To ameliorate these issues, there has been extensive research done within nanomedicine to improve drug delivery to the posterior eye in a targeted, prolonged manner. A summary of these studies is presented in Table 1.

### 5.1. Retinitis Pigmentosa

Retinitis pigmentosa (RP) is a group of inherited disorders that affect the retina. It is caused by various genetic mutations, leading to the degeneration of the photoreceptor cells, primarily rods rather than cones, and subsequent progressive vision loss beginning with night blindness and peripheral vision loss. As RP is a genetic condition with over 3000 known mutations that target specific systems or proteins, which are affected by multiple mutations, it is an effective approach to maximizing the therapeutic effect in a large patient population. Successful nano-based DDSs of therapies, which have shown success in this front, thus hold great promise. Several studies have instead focused on targeting the posterior eye to prevent retinal degeneration, such as preventing photoreceptor death and promoting its survival. This is typically accomplished by administering neuroprotective agents to retinal cells. These agents have neurotrophic, anti-apoptotic, or antioxidant properties aimed at reducing retinal inflammation, decreasing oxidative stress, and promoting repair of damaged neurons and cells [70].

Arranz–Romera et al. used PLGA microspheres to co-deliver the growth-derived neurotrophic factor (GDNF) to promote neuronal survival, with tauroursodeoxycholic acid (TUDCA), a substance shown to have anti-apoptotic, antioxidant, anti-inflammatory, and cytoprotective attributes in retinal degeneration models [71] The biodegradable nature of this microsphere allowed for a sustained, erosion-driven controlled drug release in the target tissue at effective concentrations. Through the optimization of drug loading, they were able to improve TUDCA entrapment while reducing the initial burst effect of GDNF. They observed a sustained release for at least 91 days in vitro, an essential component for RP since it requires long-acting drug responses. One benefit of nano-based formulations is the possibility of small-scale modifications that have a significant impact on the final behaviour of the DDSs and drugs. In this study, the addition of vitamin E during microsphere formulations allowed for a greater stability of GDNF during the emulsion, translating to improved GDNF function and prolonged release in vitro. Furthermore, the use of the water-soluble ethanol (EtOH) as a co-solvent-affected DDS solidification and microsphere porosity and structure, contributing to improved encapsulation efficiency of both GDNF and TUDCA. The external morphology of microparticles, modified through the addition of EtOH and other substances, affects the release profile of their encapsulated proteins [72]. Finally, combination therapy holds its own benefits, and in this experiment, it was observed that the presence of amphiphilic TUDCA modulated the release of hydrophilic GDNF. Another substance found to attenuate retinal degeneration is ML240, which inhibits the valosin-containing protein (VCP), a potential therapeutic target for autosomal-dominant RP [73]. To improve solubility and thereby maximize ML240’s therapeutic potential, Sen et al. used methoxy-poly (ethylene glycol) (mPEG)-cholane and mPEG-cholesterol-based nanoparticles that self-assemble to encapsulate the drug and improve its retention time [74]. The formulations prolonged the drug release over 10 days, and neuroprotection, particularly photoreceptor protection, was observed for up to 21 days in retinal explants with decreased inflammatory microglial responses in an ex vivo rat model. It was also observed that the formulations were safe and well-tolerated in in vivo wild-type rat eyes. However, this may not translate to rats or humans with RP as there are secondary insults and biological changes that are not present in wild-type counterparts. The study nevertheless highlights the significant role of nano-based DDSs for making accessible therapeutic targets that have shown an initial promise but are limited by their delivery and behaviour in vivo without support. Furthermore, they observed small particle sizes of mPEG-loaded nanoparticles, ranging from 32 to 55 nm, which is optimal for corneal penetration, absorption, reduced eye irritation, and patient compliance as this requires smaller needles. However, as with the above study, in vivo work is required to concretely establish therapeutic success as many initially promising therapies fail to instigate the desired effect in the complicated in vivo system. It should also be noted that neither study directly assessed the biodegradability of its proposed DDS. While both PLGA and PEG are biocompatible and degradable, it is worth exploring the biodegradability, and subsequent long-term effects of the degraded components, for specific formulations. Prioritizing patient comfort, Platania et al. developed a novel topical formulation of myriocin-loaded nanostructured lipid carriers (Myr-NLCs) in the form of eye-drops [75]. They observed that this system considerably decreased retinal sphingolipid levels in rabbit eyes, showing potential in the treatment of RP by inhibiting ceramide synthesis. The researchers observed that the Myr-NLC formulation is well-tolerated after delivery and indicated effective levels of myriocin in the posterior eye. In previous work, myriocin has shown promise in lowering retinal ceremide levels in RP mouse models when loaded in solid lipid nanocarriers (SLNs) [76]. This current work went one step further to highlight the superiority of NLCs over SLNs, particularly for drug solubility and, thus, loading. SLNs face challenges with long-term storage as there is a high chance of drug expulsion that can be overcome with NLCs, allowing for possible large-scale production if clinical success is achieved [69]. However, it is currently unclear how well these NLCs translate to in vivo efficacy. In particular, myriocin has limited stability at temperatures above 0 °C and, despite the increased stability afforded by the NLC system, it is unclear how the drug will respond at physiological temperature.

Due to the limited number of studies, and the high heterogeneity in the type and formulation of the DDSs and the active substances assessed, it is currently unclear which nano-based DDSs are most effective for RP. The longest sustained release, with a reduced initial burst release, was observed with the use of PLGA microspheres. However, whether longer release time necessarily correlates to drug efficacy and disease treatment is unclear. Regardless, it can be concluded that DDSs, which successfully enhance residence time and the stability of potential therapeutic targets that have been previously limited in their use and modulate neuroprotective effects in the retina, are likely to show the most promise in clinical applications. Overall, all studies mentioned above conducted preliminary ex vivo, in vitro, and in vivo experiments. Therefore, there is a need for in-vivo models on bigger rodents with similar anatomy to the human eye to further elucidate the therapeutic efficacy in a way that can be clinically translatable for RP.

### 5.2. Age-Related Macular Degeneration and Choroidal Neovascularization

Age-related macular degeneration (AMD) is a prevalent eye disorder that affects individuals over the age of 50 and is a major contributor to vision loss and blindness among the elderly. The condition affects the macula and results in difficulties with tasks such as reading and facial recognition. AMD can be classified into two types: dry and wet. The dry form is the most common type and progresses gradually over time. The wet form, while less common, is more severe and results from the growth of abnormal blood vessels under the macula, which then leak fluid and blood, leading to a rapid decline in vision. There are various delivery targets for AMD, including reducing inflammation and drusen formations, improving RPE survival, inhibiting angiogenesis, as well as treating choroidal neovascularization (CNV) found in wet-AMD. In a clinical setting, the treatment for AMD depends on its severity and type. Dry AMD can be monitored and managed with nutritional supplements, while wet AMD typically requires regular intravitreal injections of anti-VEGF drugs [77,78].

Anti-VEGF therapy has been one of the most common therapies for treating wet-AMD and CNV, and nano-based DDS systems to improve its delivery will be extensively reviewed in the next sections. However, one-third of patients respond poorly to anti-VEGF based treatments, and there are potential vision-threatening complications such as endophthalmitis or retinal detachment. Intravitreal injections of anti-VEGF also heavily rely on a patient’s compliance [79,80]. Therefore, there is a need for optimizing therapies targeting the inflammation, degeneration, and development of the neovascularisation.

There have been efforts in creating biomimetic nano-based DDSs to improve targeted delivery to CNV lesion sites in the eyes of AMD patients. Zhang et al. used mesenchymal stem cells (MSCs) to carry PLGA nanoparticles loaded with HIF-1α siRNA. Inhibiting HIF-1α can reduce a variety of pro-angiogenic factors working upstream of VEGF [81].

Given that hypoxia plays a major role in the pathogenesis of CNV, the study was conducted within a hypoxic environment. MSCs were able to target CNV lesion sites in this environment with the biodegradable nanoparticles improving the drug-carrying capacity and sustained release. Drug delivery through stem cell loading reached clinical trials in several cases, including apoptotic-inducing factors and oncolytic viruses, holding promise for MSC-guided delivery in retinal disorders [82]. Here, a compounded benefit is observed in which combination therapy overcomes the individual barriers to each component. siRNA alone is prone to RNAse enzymatic degradation, but encapsulation in PLGA NPs has proven protective for siRNA. Likewise, MSCs alone have poor drug carrier capacity due to poor drug loading, which can be ameliorated with the engineering of MSCs with NPs, enhancing drug loading and therapeutic efficacy. The PLGA NPs-loaded HIF siRNA effectively decreases expression of HIF-1α for 7 days in retinal pigment epithelial (RPE) cells. However, no significant difference was observed in the proliferation, apoptosis, or migration of RPEs when compared to control groups, suggesting that more work is needed to characterize how well MSC-guided delivery translates to physiological and functional improvement. Overall, this formulation requires further optimization and safety testing on animals to ensure a therapeutic benefit for AMD and CNV.

To treat CNV intravenously, Xia et al. provide another biomimetic DDS using macrophages to disguise PLGA nanoparticles loaded with rapamycin [77]. Rapamycin is an mTOR inhibitor that is known to suppress inflammation, enhance the dysregulated autophagy observed in AMD, and act upstream to VEGF-mediated inhibition of angiogenesis. Although a promising therapeutic drug for AMD, rapamycin’s low water solubility and poor accumulation at lesion sites have historically limited its use. Using the knowledge that macrophages are generally recruited to areas of RPE atrophy and CNV lesions, Xia et al. applied this to deliver PLGA-rapamycin nanoparticles intravenously in a laser-induced CNV mouse model. PLGA, as a hydrophobic drug carrier, opens the door to several potential drugs with limited water solubility despite an initial promise. The drug successfully traversed the impaired BRB, improved the bioavailability of rapamycin, and, along with anti-angiogenic effects, contributed to suppressed neovascularization. Rapamycin delivery also suppressed inflammation and enhanced autophagy both in vitro and in vivo in a CNV mouse model. Xia et al. carefully parsed out the mechanisms of action of macrophage-guided drug delivery and subsequent impact on the retinal microenvironment successfully, and characterized both in vitro and in vivo behaviour, paving the way for future clinical work to characterize the use of this formulation more effectively in humans. Using biomimetic carriers could, therefore, provide an alternative way to improve posterior ocular delivery. Rapamycin was also delivered intravitreally using synthetic high-density lipoprotein (sHDL) nanoparticles in a study by Mei et al. [78]. They particularly focused on a treatment for dry AMD, using rapamycin to suppress inflammation through the inhibition of NF-κB, as well as enhance autophagy, and using sHDL to also reduce lipid deposition, contributing to drusen formation. This DDS altogether provided a non-toxic, synergistic, anti-inflammatory effect and improved the bioavailability and distribution of rapamycin to the RPE layer following intravitreal administration in rats, with as much as a 125-fold increase in drug aqueous concentration. Combined with the observed benefits of macrophage-guided rapamycin delivery, it can be said that rapamycin is a promising drug for AMD and CNV, both because of its influence on VEGF production as well as the general effects on apoptosis, autophagy, and inflammation. This study also highlights the benefits of combined therapy, as sHDL itself had protective effects through the removal of excess cholesterol alongside its role as the nanocarrier. It also circles back to the influence of nanocarriers in effectively delivering hydrophobic drugs in largely hydrophilic environments, such as the ocular environment. However, it should be noted that neither study exploring rapamycin efficacy has explored the longevity of their formulations and the effects of long-term delivery of rapamycin in the posterior eye segment. Further studies using disease animal models are also needed to validate therapeutic efficacy and modify these therapies for clinical translation. Moreover, there are adverse side effects associated with frequent intravitreal injections.

Oxidative stress and the production of reactive oxidative species (ROS) have also been implicated in the pathophysiology and progression of AMD, thus targeting ROS production to initiate antioxidative effects. To explore this, Nguyen et al. intravitreally co-delivered resveratrol and metformin using poly(ε-caprolactone) (PCL) nanoparticles as a potential therapy for wet AMD [83]. Combined with metformin’s anti-angiogenic effects, resveratrol has been noted to provide antioxidant and anti-inflammatory effects. Due to the multifaceted effects of ROS-initiated RPE damage, therapies that can simultaneously target several components at once are highly desirable. The advantages of PCL, including its biodegradability, are mentioned, where PCL is not only considered more biocompatible in the RPE regions, but its degraded by-products are less acidic when compared to PLGA and PLA, which result in the build-up of lactic acid, avoiding unnecessary associated inflammation. It is also FDA-approved, thus easing progression in clinical trials. The polymer was further modified with cell-penetrating peptides (CPPs) to significantly improve retinal permeability. A sustained release for up to 56 days, as well as therapeutic effects, were observed in a rat model of AMD. This study provides a foundation for future long-term efficacy and safety studies. Another co-delivery system was suggested by Lai et al. for berberine hydrochloride and chrysophanol, which possesses potent antioxidant, anti-angiogenic, and anti-inflammatory properties [54]. These drugs have demonstrated potential in the treatment of AMD in animal studies. Previously limited in their application due to poor stability and bioavailability, Lai et al. proposed using polyamidoamine dendrimers (PAMAM) and liposomes to effectively deliver berberine hydrochloride and chrysophanol to the retina. PAMAM acts as an external coating for the compound-loaded liposomes due to its high water-binding ability and low toxicity. In comparison to uncoated compound liposomes, this coated DDS revealed a negative zeta potential, which is preferred for drug delivery to the retina, and significantly improved encapsulation efficiency, demonstrating that PAMAM coating enhanced drug loading. Results show considerable cellular permeability and increased bio-adhesion on corneal epithelial cells. PAMAM-liposome systems (P-CBLs) also substantially improved berberine hydrochloride bioavailability. Further, no side effects were observed on rabbit ocular surface structure after the administration of P-CBLs. While the drugs exhibited stability for 7 h in vivo, the study did not assess the release profiles of the drugs in the posterior segment of the eye, leaving questions regarding the functionality of this DDS in AMD. Regardless, the P-CBL system displays a potential use for treating AMD and, potentially, other ocular diseases.

Oxidative stress and the production of reactive oxidative species (ROS) have also been implicated in the pathophysiology and progression of AMD. Thus, targeting ROS production to initiate antioxidative effects. To explore this, Nguyen et al. intravitreally co-delivered resveratrol and metformin using poly(ε-caprolactone) (PCL) nanoparticles as a potential therapy for wet AMD [83]. Combined with metformin’s anti-angiogenic effects, resveratrol has been noted to provide antioxidant and anti-inflammatory effects. Due to the multifaceted effects of ROS-initiated RPE damage, therapies that can simultaneously target several components at once are highly desirable. The advantages of PCL, including its biodegradability, are mentioned, where PCL is not only considered more biocompatible in the RPE regions, but its degraded by-products are less acidic when compared to PLGA and PLA, which result in build-up of lactic acid, avoiding unnecessary associated inflammation. It is also FDA-approved, thus easing progression in clinical trials. The polymer was further modified with cell-penetrating peptides (CPPs) to significantly improve retinal permeability. A sustained release for up to 56 days, as well as therapeutic effects, were observed in a rat model of AMD. This study provides a foundation for future long-term efficacy and safety studies. Another co-delivery system was suggested by Lai et al. for berberine hydrochloride and chrysophanol, which possesses potent antioxidant, anti-angiogenic, and anti-inflammatory properties [54]. These drugs have demonstrated potential in the treatment of AMD in animal studies. Previously limited in their application due to poor stability and bioavailability, Lai et al. proposed using polyamidoamine dendrimers (PAMAM) and liposomes to effectively deliver berberine hydrochloride and chrysophanol to the retina. PAMAM acts as an external coating for the compound-loaded liposomes due to its high water-binding ability and low toxicity. In comparison to uncoated compound liposomes, this coated DDS revealed a negative zeta potential, which is preferred for drug delivery to the retina and significantly improved encapsulation efficiency, demonstrating that PAMAM coating enhanced drug loading. Results show considerable cellular permeability and increased bio-adhesion on corneal epithelial cells. PAMAM-liposome systems (P-CBLs) also substantially improved berberine hydrochloride bioavailability. Further, no side effects were observed on the rabbit ocular surface structure after the administration of P-CBLs. While the drugs exhibited stability for 7 h in vivo, the study did not assess the release profiles of the drugs in the posterior segment of the eye, leaving questions regarding the functionality of this DDS in AMD. Regardless, the P-CBL system displays a potential use for treating AMD and, potentially, other ocular diseases.

### 5.3. Diabetic Retinopathy

Diabetic retinopathy is a chronic ocular condition affecting diabetic patients. The condition results from damage to the blood vessels in the retina and can progress over time. There are two main stages of diabetic retinopathy: non-proliferative diabetic retinopathy (NPDR) and proliferative diabetic retinopathy (PDR). NPDR is characterized by increased vascular permeability and capillary occlusion, and can lead to the formation of microaneurysms, dot and blot hemorrhages, cotton wool spots, and hard exudates. PDR occurs in advanced stages of diabetic retinopathy due to continued damage to the retinal blood vessels, leading to significant retinal ischemia. The ischemic retinal tissue releases pro-angiogenic factors, including the vascular endothelial growth factor (VEGF), which stimulates the production of new and abnormal blood vessels. These neo-vessels can lead to various vision-threatening complications, such as a neovascularization of the disc and retina causing vitreous hemorrhage and tractional retinal detachment, and neovascularization of the iris and angle resulting in glaucoma. The management of Proliferative Diabetic Retinopathy primarily focuses on reducing the production of VEGF by ischemic tissue through laser photocoagulation or intravitreal anti-VEGF injections.

Antioxidants, anti-inflammatory agents, and neurotrophic factors are considered promising options to treat the neuronal and vascular abnormalities that progress with diabetic retinopathy (DR) [84]. Nano-carriers have been proposed to improve the targeting of the diabetic retina. Due to their high biocompatibility, PLGA-based nanoparticles have been used to improve the therapeutic efficacy of drugs that are currently limited due to inefficient delivery routes. For example, Zeng et al. used PLGA nanoparticles to deliver Interleukin-12 (IL-12), a cytokine that can diminish the levels of matrix metalloproteinase-9 (MMP-9) and VEGF-A, both of which are known to affect the severity of diabetic retinopathy [85]. Previously limited due to it being prone to rapid degradation, when IL-12 was carried by PLGA nanoparticles (IL-12-PNP), it had an appreciable drug encapsulation efficiency (~34.7%) and prolonged drug release. IL-12-PNP exhibited better inhibition against VEGF-A and MMP-9 expression in diabetic retinopathic mouse retina and rat endothelial cells. Moreover, this treatment resulted in significantly decreased retinal damage in a DR mouse model with increased thickness and reduced neovascularization. Similarly, Romeo et al. proposed to deliver melatonin with PLGA-PEG Lipid-polymer hybrid nanoparticles (LPHN) [86]. Melatonin offers various neuroprotective strategies suitable for treating this DR. However, at high doses, it may compromise retina morphology and functioning. The DDS developed in this study targeted the retina without unnecessary high dosages to deliver melatonin. Using a biodegradable polymer, they found no signs of cytotoxicity or ocular irritation in vivo and confirmed neuroprotective and antioxidant effects on a model of glucose-induced diabetic retinopathy on Human Retinal Endothelial Cells (HREC). They also observed high encapsulation efficacy (79.8%) using this hybrid model, suggesting its superiority to a PLGA only nanoparticle. In previous work, the neuroprotective effects of melatonin have been observed only after prolonged exposure of greater than 72 h, necessitating a stable, sustained release DDS for its ocular delivery. Romeo et al. successfully observed a prolonged and sustained release for up to 8 days compared to a rapid burst release of free melatonin.

Another example of a lipid-modified nanoparticle system is a study by Zingale et al., where they used nanostructured lipid carriers (NLCs) to deliver diosmin, a flavonoid known for its anti-inflammatory, cytoprotective, and antioxidant effects, especially in high glucose environments [87]. They were able to achieve a high encapsulation efficiency, and the DDS was found safe and well-tolerated in vitro. However, a common issue observed with using lipid-based nanocarriers is the need to use surfactants for their preparation that may cause irritation and a sensitizing action [88]. Further studies are being conducted to confirm the clinically relevant anti-inflammatory effects of diosmin NLCs. As mentioned above, NLCs have the advantage of minimal toxicity as it can be manufactured without the requirement of toxic organic solvents [89]. They can also be stored stably for long periods, as Zingale et al. observed stability under different storage conditions for up to 60 days. NLCs further possess the versatility of being applied as topical eye drops as demonstrated here and also by Platania et al., which greatly increases patient compliance [74]. What’s currently unclear and garners further exploration is the release profiles of drug-loaded NLCs, to better assess how often administration is required.

Other types of biodegradable nanoparticles have also been assessed for optimizing treatments for diabetic retinopathy. Radwan et al. investigated an alternative non-invasive delivery with an anti-VEGF factor, apatinib, encapsulated into bovine serum albumin nanoparticles, which are coated with hyaluronic acid [90]. With a relatively high entrapment efficiency (~69%), these apatinib-loaded nanoparticles (Apa-HA-BSA-NPs) illustrated a sustained biphasic release rate with an initial burst, appreciable mucoadhesion, and no cytotoxicity were detected on rabbit corneal epithelial cells. This 2021 study indicated improved retinal thickness and lessen retinal microstructural and ultrastructural changes in Apa–HA–BSA–NP-treated eyes. Moreover, the authors observed better retinal accumulation through this topical treatment while avoiding ocular complications resulting from frequent intravitreal injections. As aforementioned in the AMD section, using PCL as a biodegradable polymer for nanoparticle systems has many advantages [83]. For diabetic retinopathy, Mahaling et al. developed nanoparticles with a hydrophobic polycaprolactone (PCL) core and a hydrophilic Pluronic^®^ F68 shell, containing triamcinolone acetonide [91]. TA has demonstrated efficacy in both NPDR and PDR, attributed to its anti-inflammatory, anti-angiogenic, and neuroprotective properties. Likewise, NPs containing PCL and PF68 have previously demonstrated strong bioavailability in retina during topical administration [92]. In a DR rat model, a topical administration of these nanoparticles resulted in significant structural improvements, particularly retinal thickness and vascular health, as well as functional improvements. The authors found diminished retinal inflammation, decreased glial cell hyperplasia, and reduced microvascular complications. These findings demonstrate the potential of a triamcinolone acetonide-loaded nanoparticle delivery system in the treatment of diabetic retinopathy. Topical administration has observed significant success in DR animal models, opening the door to non-invasive, patient self-administered delivery routes. This overcomes several challenges of intravitreal administration, including intraocular bleeding, increased intraocular pressure, endophthalmitis, and discomfort.

### 5.4. Diabetic Macular Edema (DME)

Diabetic macular edema (DME) is a common complication in diabetic retinopathy where fluid accumulates in the macula causing rapidly progressive decrease in visual acuity. It occurs due to increased permeability and inflammation in the retinal vessels [84].

Other than intravitreal anti-VEGF injection and topical NSAIDs, intravitreal triamcinolone acetonide (TA) can sometimes be used to reduce associated inflammation with DME. However, intravitreal triamcinolone is associated with excessively high rates of complications, such as IOP elevation and cataract formation. Navarro–Partida et al. provided a topical route for delivering TA by loading it on liposomes [93]. This was a feasibility study, where they first found TA-loaded liposomes to be safe and tolerable in healthy patients through a Phase 1 clinical assay. They further presented a sustained therapeutic effect of reduced central fovea thickness (CFT) in DME patients through an open-label, non-randomized study. Further studies are required to confirm the long-term safety and therapeutic efficacy, such as ensuring TA at high concentrations does not adversely affect intraocular morphology and function [94]. To improve the biodegradability and mucoadhesion of liposomes, Khalil et al. used chitosan-coated liposomes to deliver TA to the posterior segment [95]. This enhanced bioavailability and prolonged the release of TA in their in vivo models. Although their efficiency of drug release was done on a CNV rat model, the authors recommend this DDS for any posterior segment disease, particularly highlighting DME, proliferative diabetic retinopathy, and CNV related to AMD. Further in vivo studies are required to validate the therapeutic efficacy of this DDS, ensuring its clinical significance. Initial clinical success with TA topical administration in lipid-based nanomaterial has further supported both the superiority of lipid-based DDSs and topical administration in ocular drug delivery. Khalil et al. further demonstrate the flexibility afforded by nano-based DDS, as base constructs, such as liposomes, can be modified to improve retention time, permeability, encapsulation efficiency, and personalize treatment to the drug, disease, and area of interest.

**Table 1 pharmaceutics-15-01094-t001:** Biodegradable DDS for posterior segment diseases.

Disease	DDS Technology	Drug	Advantages & Considerations	Administration Route	Stage	Reference
RP	Self-assembled PEG-based NPs	ML240 (VCP inhibitor)	Prolonged drug release Long-lasting neuroprotective effect	IVT injection	Preclinical: ex vivo, in vitro, in vivo	[74]
	NPs	Myriocin	Effective level of myriocin at back of eye Decrease retinal sphingolipid levels	Topical	Preclinical	[75]
	PLGA Microspheres	GDNF and TUDCA	Sustained dual drug release Neuroprotective, cytoprotective effects Preliminary study, further confirmatory studies required	IVT injection	Preclinical: in vitro	[71]
	PVA/PVP/PG polymer	Progesterone	Good biocompatibility, controlled release Accumulates in sclera Delays photoreceptor cell death Further studies on therapeutic efficacy required	Ocular inserts	Preclinical: in vitro, ex vivo	[96]
Wet AMD	PCL NPs	Resveratrol and Metformin	Enhanced retinal permeability Combined anti-inflammatory, antioxidant, and anti-angiogenic effects	IVT injection	Preclinical: in vitro and in vivo	[83]
CNV	MSC-transfected PLGA NPs	HIF-1α siRNA	Reduced HIF-1α activity in hypoxic environment Biomimetic delivery system Preliminary validation study	IVT injection	Preclinical: in vitro	[81]
Dry and wet-AMD	sHDL NPs	Rapamycin	High encapsulation efficiency Dual function of reducing cholesterol in tissue targeted Targeted anti-inflammatory effects on RPE	IVT injection	Preclinical: in vivo	[78]
	Solid lipid NPs	Atorvastatin	Prolonged residence More bioavailability Improved stability	Topical	Preclinical	[97]
	Nanoceria	Glycol Chitosan	Decrease ROS-induced pro-angiogenic VEGF	IVT injection	Preclinical	[98]
	PAMAM-coated liposomes	BBH and Chrysophanol	Appreciable cellular permeability Improved BBH bioavailability	Topical	Preclinical	[54]
Wet-AMD/ CNV	MRaNPs	Rapamycin	Biomimetic non-invasive DDS Improved accumulation in CNV lesions Anti-angiogenic, anti-inflammatory, enhanced autophagy effects	IVT injection	Preclinical: in vivo	[77]
	Porous poly (PDMS) capsule	Ranibizumab	Sustained released for 16 weeks Reduced CNV area	Transscleral	Preclinical	[99]
PDR/ Wet-AMD	NPs	Fenofibrate	Prolonged drug release Beneficial effect on neovascular AMD No toxicity detected	IVT injection	Preclinical	[100]
PDR	NPs coated with HA	Apatinib	Show size, Pdl and zeta potential High entrapment efficiency	Topical	Preclinical	[90]
	Nanoparticles	Interleukin-12	Sustained drug release Effective drug treatment Restore thickness	IVT injection	Preclinical	[85]
	NPs	Triamcinolone acetonide	Non-invasive delivery Improve structural and functional activity Reduce retinal inflammation and vascular abnormalities	Topical	Preclinical	[91]
	PLGA-PEG Lipid-polymer hybrid NPs	Melatonin	Confirmed in-vitro antioxidant and neuroprotective effectiveness Confirmed ocular tolerability, no cytotoxicity in vivo	Topical	Preclinical: in vitro and in vivo	[86]
	Nanostructured lipid carriers	Diosmin	Cytoprotective, anti-inflammatory effects Confirmed ocular tolerability and safety in vitro Preliminary study, studies regarding therapeutic efficacy further required	Topical	Preclinical: in vitro	[87]
Wet-AMD/CNV/DME/ PDR	Chitosan coated liposomes	Triamcinolone Acetonide	Increased bioavailability Sustained drug release Improved biodegradability and mucoadhesion	Topical	Preclinical: in vivo	[96]
DME	Liposomes	Triamcinolone Aceonide	Sustained release Confirmed safety, tolerability, and therapeutic activity in humans Further long-term safety and therapeutic efficacy clinical studies required	Topical	Clinical: in vivo, Phase 1 clinical assay	[93]

RP: retinitis pigmentosa; AMD: age-related macular degeneration; CNV: Choroidal neovascularization; PDR: proliferative diabetic retinopathy; DME: diabetic macular edema; PEG: polyethylene glycol; NPs: nanoparticles; PLGA: poly-lactic-co-glycolic acid; PVA: polyvinyl alcohol; PVP: polyvinylpyrrolidone; PCL: polycaprolactone; MSC: mesenchymal stem cells; sHDL: synthetic high density lipoprotein; PAMAM: poly(amidoamine); MRaNPs: macrophage-disguised nanoparticles; HA: hyaluronic acid; VCP: vasolin-containing protein; GDNF: glial cell derived neurotrophic factor; TUDCA: tauroursodeoxycholic acid; RPE: retinal pigmented epithelium; BBH: berberine hydrochloride; IVT: intravitreal; DDS: drug delivery systems; ML240: Mycobacterium leprae 240; PG: prostaglandin; HIF-1α: hypoxia-inducible factor 1-alpha; siRNA: small interfering RNA; ROS: reactive oxygen species; VEGF: vascular endothelial growth factor; PDMS: polydimethylsiloxane; Pdl: polydispersity index.

## 6. Anti-VEGF Agents

As highlighted in the previous section, pathological neovascularization plays a role in the underlying mechanism of various retinal diseases. This type of neovascularization is caused by retinal hypoxia and is involved in the pathophysiology of several retinal disorders, including proliferative diabetic retinopathy (PDR), retinopathy of prematurity (RoP), and retinal vein occlusion (RVO) [101,102]. On the other hand, CNV often occurs due to a ruptured or damaged Bruch’s membrane, which can be caused by various retinal disorders such as wet age-related macular degeneration, pathologic myopia, presumed ocular histoplasmosis syndrome (POHS), and traumatic choroidal rupture [103]. Anti-VEGF agents have emerged as the gold standard treatment to treat ocular neovascular diseases. Commonly used anti-VEGF drugs include Bevacizumab (Avastin™), Ranibizumab (Lucentis™), Aflibercept (Eylea™), and Pegaptanib (Macugen^®^). These drugs, which are monoclonal antibodies that target retinal and choroidal endothelial cells to stop angiogenesis, face a challenge in terms of bioavailability and targeted delivery. This is due to their large size, which makes it difficult for them to penetrate through ocular barriers such as the blood–aqueous barrier and blood–retinal barrier. Anti-VEGF agents have a short half-life and, thus, need to be administered regularly by intravitreal injections. The latter is associated with potential sight-threatening complications such as endophthalmitis and retinal detachment [104].

The use of non-degradable implants for anti-VEGF delivery has been proposed, but it has some drawbacks. While non-degradable implants can provide a long-term drug release, they require secondary surgery to remove the depleted material, which is associated with additional risks and potential complications. Moreover, these non-degradable implants have the potential to cause several issues. The large size of the non-biodegradable implant may impact the visual axis. There is also a risk of implant migration to the anterior chamber, which may lead to corneal edema and permanent endothelial decompensation caused by direct contact with the endothelium, mechanical trauma, or chemical toxicity [105].

The use of biodegradable nanocarriers minimizes the adverse effects associated with non-degradable implants containing anti-VEGF. These biodegradable, anti-VEGF sustained-release drugs utilize biopolymers to deliver the drug for sustained release. The carrier material is gradually degraded within the eye, eliminating the need for a second surgery, and the release rate can be adjusted by modifying the composition and molecular weight of the carrier. Furthermore, their small size allows for optical clarity and reduces the risk of visual disturbances [105]. In summary, biodegradable nanocarriers present, potentially, a favorable option for enhancing the efficacy, bioavailability, bioactivity, duration of action, and safety of anti-VEGF treatment. Table 2 showcases a comprehensive overview of the latest and most pertinent studies that have been published, offering valuable insights and key findings.

The most used anti-VEGF drug in ocular DDS is bevacizumab, followed by aflibercept and ranibizumab [106]. Bevacizumab is the most used in the following studies, especially with wet-AMD, as it is also one of the oldest anti-VEGF drugs [107]. Therefore, its toxicity and pharmacokinetic characteristics are well-known. Aflibercept and ranibizumab are also used for treating wet-AMD. Aflibercept is the most potent drug and has been shown to require less frequent dosing due to its longer duration of action, which can be advantageous in terms of patient adherence. However, there are also some concerns about the rare but severe adverse effects related to its use [108]. It is also much more costly compared to bevacizumab. This may explain why it is not the most common anti-VEGF drug in the current clinical practice.

### 6.1. Novel DDS for Anti-VEGF Agents

Nanocarriers can be classified according to their material components: lipid-based, polymers and inorganic nanoparticles. Each class has its advantages and challenges. Anti-VEGF agents are sensitive to conformational changes, and their stability can be easily compromised by in vivo triggers. It remains a challenge to protect the drugs from protein denaturation while minimizing the interaction with the nanocarrier to preserve the drug’s bioactivity. If interactions are too strong, they can compromise drug capture and release processes [109].

#### 6.1.1. Hydrogel

Hydrogel nanocarriers are three-dimensional polymer networks with porous structure. The polymers are hydrophilic and can, thus, interact with molecules that have a high-water solubility. Hydrogels possess the unique feature of carrying water molecules yet remain under a solid state [110]. Hydrogel emerged as a top nanocarrier choice for its excellent biocompatibility, biodegradability, and safety profile [69]. They are known to have a rapid high initial burst whereby 10–50% of the drug can be lost through diffusion [110]. Sterilization processes can affect their delicate structure [111].

Some hydrogels are referred to as “smart” because they can change their properties in response to environmental triggers, such as pH or temperature changes [112]. Osswald et al. previously developed a PNIPAAm–PEG–diacrylate thermoresponsive hydrogel composed of suspended PLGA microspheres to carry ranibizumab and aflibercept [113]. They discovered that by suspending the microspheres in the hydrogel, the drug release was extended by 27.2%. Therefore, the nano-based DDS successfully released ranibizumab, or aflibercept, for 196 days while remaining bioactive in vitro. Promising findings were obtained in vitro as the DDS inhibited human umbilical vein endothelial cell (HUVEC) proliferation. It, thus, encouraged the team to pursue experiments on in vivo models. Later in 2017, Osswald et al. published results on laser-induced rat CNV models [114]. The nanotherapeutic significantly reduced CNV lesion areas by 60% compared to the control group in vivo. Over the course of 12-week treatment, less drugs were needed in the novel nanotherapeutics compared to the standard posology delivered via bolus administration. While this DDS could become advantageous compared to the standard treatment by limiting toxicity related to high drug dosage, it is important to highlight the small animal samples per treatment group, which was of four eyes. The results are, thus, limited.

Similarly, Hu et al. tested bevacizumab in a synthesized thermoresponsive mPEG–PLGA–BOX hydrogel [115]. The hydrogel transitioned from a solution phase to a gel-phase after body temperature exposition. Both in vitro and in Rex rabbits, the nanotherapeutic inhibited angiogenesis induced by retinal laser photocoagulation over the course of 35 days. After intravitreal injection, the anti-angiogenic bioactivity of bevacizumab was maintained. No cytotoxic effects during the nanocarrier biodegradation were reported in Rex rabbits. This experiment was conducted on 11 Rex rabbits divided in two groups. Given this small animal sample, results remain preliminary. However, it shows that DDS might offer promising results as a novel therapeutic gelling carrier against angiogenesis.

Xue et al. encapsulated bevacizumab and aflibercept in a thermoresponsive hydrogel synthesized with PED–PPG–PCL [116]. As expected, the hydrogel exhibited good biocompatibility and no toxicity. Tests were conducted in vitro on bevacizumab and aflibercept separately. Both drugs significantly inhibited proliferation in HUVEC. Both anti-VEGF drugs were independently injected with the nanocarrier in an ex vivo choroidal sprouting model and significantly reduced the relative sprouting percentage by more than 80% compared to the control hydrogel. Anti-angiogenic effects were reported ex vivo and in vivo on a persistent retinal neovascularization rabbit model. This confirmed sustained drug bioactivity in the nanocarrier. The hydrogel was fine-tuned by modifying the hydrophilic/lipophilic ratio to extend the prolonged drug release rate. After increasing the hydrogel concentration to 20 weight percent with the optimized PEG/PPG ratio of 4:1, the longest drug release of 40 days in vitro and of at least 28 days in vivo was obtained. The novel DDS represents a potential bioactive drug carrier with a prolonged drug release rate that can be extended via polymer fine tuning.

Thermoresponsive hydrogel studies have shown optimistic results in vitro and in vivo, but for limited periods of time ranging from days to weeks. Liu et al. also explored the use of a thermorepsonsive hydrogel to deliver bevacizumab, but over the course of 6 months in vitro. They used PGLA in a poly(ethylene glycol)-co-(L-lactic-acid) diacrylate/*N*-isopropylacrylamide (PEG–PLLA–DA/NIPAAm) thermoresponsive hydrogel loaded with ranibizumab [38]. By increasing cross-linker concentration and not charging the microsphere with more than 20 mg/mL, optimal conditions were achieved through enhanced biodegradability, drug release, and needle-injection feasibility. The hydrogel proved to be effective in vitro for 190 days. Liu et al. then pursued to test the novel DDS with aflibercept in vitro [117]. Aflibercept was successfully released for 6 months while maintaining bioactive therapeutic levels. Drug quantity and release could be tuned based on cross-linker PEG–PLLA–DA concentration and microsphere load quantity. The biodegradable cross-linker PEG–PLLA–DA prolonged the hydrogel nanocarrier degradation. Liu et al. then proceeded to inject intravitreally the aflibercept-DDS in a laser-induced CNV rat model [118]. The nanotherapeutic was as effective as a bimonthly aflibercept injection to treat CNV lesion areas for 6 consecutive months while avoiding inflammation and ocular complications. This nanotherapeutic proved to be safe and biocompatible in vivo on the rodent eye model. An important limitation of this promising DDS is its potential non-applicability to humans given the anatomical differences between the rodent and human eyes. Therefore, the drug pharmacokinetics and DDS-related immune reactions may differ.

Fan et al. developed a short chain peptide to deliver conbercept, a novel anti-VEGF drug with a short half-life, in vitro [119]. The peptide was pH-sensitive and self-assembled as a hydrogel when triggered by a pH under 7.4. The nano-based DDS inhibited the proliferation and tube formation of human retinal endothelial cells (HREC), which suggests its potential therapeutic avenue for neovascular AMD. The hydrogel peptide nanocarrier did not affect the viability of human retinal endothelial cells (HRECs), which shows its biocompatibility. However, given that the DDS was not tested in vivo, the pharmacokinetics of the DDS remain theoretical. Results are too preliminary to further comment.

Hydrogels can be combined with different materials to gain new properties. In this recent study, Li et al. injected co-loaded sunitinib and acriflavine liposomes in a hydrogel (cSA@Lip-HAC) [120]. Acriflavine inhibits the hypoxia-inducible factor, while sunitinib acts as an anti-VEGF agent. In vitro results demonstrated that the nanocarrier exhibited high antiangiogenic properties. The increased antiangiogenic effect was enabled by the combination of co-drug-loaded liposomes in the injectable hydrogen and the chosen sub-tenon administration route. Increased retinal and choroid drug residency were reported, as well as significant anti-CNV results. The drug was able to remain 21 days in the nanocarrier in vivo. Impressively, the nanotherapeutic showed increased anti-CNV efficacy in the laser-induced CNV rat models when compared to an intravitreal commercial conbercept injection. This DDS thus represents a promising novel therapeutic avenue with less complications than with the intravitreal administrative route. The drug was able to remain 21 days in the nanocarrier in vivo, which is a good preliminary result. Nonetheless, the DDS remains limited in time. To ensure patient compliance, the DDS should ideally remain active for months in the eye.

#### 6.1.2. Polymers Nanoparticles and Microparticles (MPs)

In recent years, polymers have been the most tested DDS with anti-VEGF drugs. This is due to their high versatility and tuning properties. They can encapsulate various types of hydrophilic and hydrophobic molecules ranging from peptides to biological macromolecules. Polymers are widely studied as their drug release rate and other pharmacokinetic characteristics can be modified by adjusting their composition and ratios as well as combining different biomaterials. They represent promising nanocarriers due to their biodegradability and nontoxic profile. They can either be natural, synthetic or a mix of to gain desired environmental, release and retention rate features. The most studied and successful formula are bevacizumab loaded PLGA and chitosan-based nanoparticles.

PLGA-based nanocarriers are one of the most promising synthetic nanocarriers given their biocompatibility, nontoxicity, degradability, and versatility. They are FDA-approved polymers in clinical applications. They are composed of a hydrophobic core that carries hydrophobic drug and a hydrophilic outer shell (corona) that modulates drug release. Due to its amphiphilic nature, PLGA-based polymers can be used with a variety of drugs.

Tanetsugu et al. developed PLGA microspheres that delivered ranibizumab biosimilar in vitro [121]. After three weeks, more than 80% of the drug was delivered. The DDS also inhibited HUVECs tube formation. The microsphere completely degraded after 1.5 months. This DDS could become a practical system in prolonged anti-VEGF drug release treatment.

Longer drug release results were obtained by Sousa et al. who also encapsulated bevacizumab in PLGA loaded nanoparticles [122]. In vitro results showed that the DDS preserved the drug bioactivity and could deliver drugs to targeted cells in HUVEC. In 2018, Sousa et al. developed a lyophilized protocol to study the stability of encapsulated bevacizumab loaded PLGA nanoparticles [123]. They successfully stored over 6 months bevacizumab while preserving its angiogenic effect. This shows that bevacizumab could be used for prolonged release time. Similarly, Zhang et al. used PLGA nanoparticles to encapsulate bevacizumab [124]. In vitro results showed that the nanotherapeutic was more efficient than bevacizumab alone to inhibit HUVEC proliferation and tube formation. In vivo experiments on oxygen-induced retinopathy (OIR) mouse models showed that the nanotherapeutic increased the drug bioactivity in inhibiting CNV and RNV angiogenesis. No toxicity or cytotoxicity was reported in vitro and in vivo. Therefore, this PLGA drug loaded nanoparticles could become a safe in vivo treatment.

Other than bevacizumab, few other anti-VEGF drugs were tested. Kelly et al. tested in vitro aflibercept encapsulated PLGA nanoparticles [125]. 75% of the drug was released on day seven with the DDS, compared to 100% drug release after 24 h following a standard aflibercept injection. Thus, the polymer exhibited prolonged drug release over seven days and high encapsulation efficacy. This nanocarrier has the potential to be a promising DDS for delivering aflibercept to treat retinal diseases.

Some studies show that nanocarriers can improve anti-CNV activity. Yan et al. developed a novel nanocarrier composed of PLGA-PEGylated magnetic nanoparticles to treat AMD [126]. The magnetic nanoparticles conferred multiple advantages to the DDS such as stability, biocompatibility, tunable surface modification and even increased antiangiogenic efficacy. The PEG-PLGA copolymer tested in vitro exhibited effective antiangiogenic activity. This DDS could become an effective ArMD therapy.

The use of multiple drugs within nanocarriers could increase this effect. Liu et al. developed a novel poly (D, L-lactide-co-glycolide) and polyethylenimine nanoparticle loaded with dexamethasone and added bevacizumab on the nanoparticle surface (eBev-DPPNs) [127]. The novel DDS conjugated with dexamethasone and bevacizumab increased efficacy of CNV inhibition. This was shown by effective inhibition of HUVECs angiogenesis and VEGF secretion. When injected intravitreally in rabbit laser models of CNV, the nanotherapeutic significantly decreased CNV leakage areas after 28 days.

While in vivo and in vitro studies have been conducted, Heljak et al. developed a computational model to predict behaviour of intravitreally injected PLGA bevacizumab loaded microspheres [128]. The model exhibited similar results than those obtained experimentally, which suggests its future use to assess and plan anti-VEGF treatments in clinical practice.

Some prolonged drug release results were obtained with PLGA-based nanocarrier. In this study, a different preparation technique was employed to protect bevacizumab stability. It was shown that the degradability of PLGA drug loaded microspheres can be compromised by the accumulation of lactic and glycolic acids that denature the drug, thus leading to complications. For this reason, Liu et al. explored the use of a polymeric blend composed of poly (d, l-lactide-co-glycolide)/poly(cyclohexane-1,4-diyl acetone dimethylene ketal) (PLGA/PCADK) to deliver bevacizumab-dextran [129]. They prepared the DDS with a solid-in-oil-in-water (S/O/W) emulsification, which limited the initial burst release to ensure progressed drug release that maintained therapeutic level. The novel polymer blend exhibited increased biocompatibility compared to PLGA alone. The DDS delivered drug over the course of a 50-day period in vitro and in vivo in a rabbit model. By extending drug release to over one month in vivo, this nanocarrier could become a potential anti-VEGF treatment.

Tsujinaka et al. used a polymer blend composed of PLGA-PEG to deliver sunitinib [130]. Impressively, after intravitreal injection, the DDS formed a depot that released drug over 6 months in a laser induced CNV mouse model. CNV was suppressed in the type II CNV mouse model over the drug release period. In a different mouse model, the nanotherapeutic reduced VEGF-induced leukostasis and nonperfusion, which suggests it could also be used in progressive DR therapy.

Other synthetic polymers have been studied. Jiang et al. developed a polydopamine (PDA) nanoparticle that encapsulated bevacizumab to treat AMD [131]. The nanocarrier alone possessed an antiangiogenic activity by reducing reactive oxygen species (ROS). When tested in vitro and on ex-vivo porcine eyes, the novel biodegradable nano-based DDS successfully delivered bevacizumab when triggered by ROS. This DDS has the potential to become a practical dual system that delivers antiangiogenic drugs while minimizing the production of ROS.

Cai et al. synthesized modified S-PEG polymers with arigine-glycine-aspartic acid (RGD) peptide (S-PEG-ICG-RGD-RBZ nanoparticles) to deliver anti-VEGF agents intravenously [132]. The nanoparticles exhibited an antiangiogenic activity in vitro. In laser induced CNV mouse models, the nanoparticles significantly decreased CNV lesion areas. Interestingly, the nanoparticles displayed good specificity by spending minimal time in the entire organism and by not accumulating in organs other than CNV areas, demonstrating the biosafety of this drug delivery system (DDS).

Natural polymers hold the advantage of degrading easily and are thus investigated in anti-VEGF nanocarriers. Topical route is less explored as there are more barriers to penetrate before reaching the posterior eye segments. While the current anti-VEGF administrative route in clinical practice is intravitreal injection, some studies assessed the topical route with human serum albumin (HSA) nanoparticles. HSA nanoparticles are easy to preparate and exhibit adhering properties to the mucosa of the corneal epithelium, which allows the drug to remain longer bioavailable. No toxicity is reported with the advantage of accommodating a variety of drug types and molecular sizes. This phenomenon is explained by the chemical bonding between the nanoparticles and mucins. No enzymatic cross-linkage was used to stabilize the nanoparticles. In fact, this would be due to the protein interactions between the drug and the albumin. Luis de Redin et al. used HSA nanoparticles to carry bevacizumab [133]. The bevacizumab-nanoparticles increased loading capacity by 13% compared to nanoparticles cross-linked with glutaraldehyde, which is a commonly used cross-linkage reagent. During the first five minutes in vitro, the initial burst release was evaluated to 35% of the loaded drug, followed by a decreased rate over the next 24 h. When delivered to rats as eye drops, the DDS was released over 4 h before being evacuated in the gastrointestinal tract, compared to HSA control group which was cleared in less than 1 h. These results indicated that the DDS could become a potential daily eye drop treatment. However, the DDS should remain longer in the eye. Luis de Redin et al. tested the model in vivo. Similarly to their previous study, they loaded bevacizumab in albumin nanoparticles to treat CNV [134]. The eye drops were applied daily for 1 week. In vivo, the drug loaded nanoparticles exhibited better antiangiogenic activity than with bevacizumab alone and thus used 2.4 times less drug quantity. The nanocarrier significantly increased bevacizumab neovascularization inhibiting efficacy in CNV rat models. Histopathological results revealed decreased fibrosis, inflammation and edema in rats treated with bevacizumab nanoparticles. These promising results suggest that this drug delivery system could be utilized as a daily eye drop therapy with reduced dosing requirements. Given that the in vivo experiment was only conducted on Wistar rats, the results lack validation on animal models that can be more easily transposed to the human eye anatomy.

Llabot et al. also used HSA, but with added Gantrez^®^ ES-425 polymer to coat bevacizumab or suramin loaded nanoparticles [135]. The nanoparticles were tested in vitro topically and developed to treat CNV. Bevacizumab released in a small initial burst and was progressively released. The stabilizing polymer Gantrez^®^ was compared with the common cross-linked reagent, glutaraldehyde. Gantrez^®^ polymer exhibited better results than glutaraldehyde in terms of drug stability and preserved bioactivity. In vitro release results showed that 80% of suramin was released within 8 h compared to 50% for suramin. In vivo studies will be carried out on animal CNV models.

Abdi et al. explored the interaction between chitosan nanoparticles and bevacizumab [55]. Study showed that chitosan and bevacizumab had low interactions between one another, thus allowing efficient capture and release of bevacizumab. This study reported that bevacizumab and chitosan could form a successful nanocarrier. As seen in other studies, the combination of this drug and nanocarrier appears to be promising.

Several studies that used chitosan nanoparticles report effective drug release for less than 1 month. Pandit et al. merged PLGA and chitosan in nanoparticles to deliver bevacizumab through a subconjunctival injection [136]. Coating PLGA nanoparticles with chitosan reduced initial drug burst release to 25% as opposed to the drug solution control which released 90% of drug content within 24 h. In vitro, the DDS extended drug residency in the retina. The drug was released sustainably over 72 h. These preliminary results show that the nanocarrier could become a suitable DDS for the subconjunctival administration route. Ugurlu et al. proceeded by a subtenon injection of loaded chitosan particles with bevacizumab in rabbits’ eyes [137]. In vitro, the DDS progressively released drug for 3 weeks. However, in vivo results decrease within one week despite better control and more progressive drug release from the DDS (6 μg/mL for DDS and 4 μg/mL for bevacizumab). In Savin et al. study, they synthesized bevacizumab loaded chitosan grafted-poly(ethylene glycol) methacrylate nanoparticles. The solubility of chitosan polymer was increased through Michael addition reaction. The nanoparticles exhibited no toxic effect and released successfully bevacizumab in vitro for an estimate time ranging from 14 to 30 days [138].

Unlike previous studies, Jiang et al. developed a chitosan-based nanocarrier that can release drugs over months. They designed a polycaprolactone (PLA) chitosan bi-layered hybrid shell capsule that delivers bevacizumab in hope to treat AMD [139]. This structure was chosen to load high drug amounts. In vitro results impressively showed that the DDS released drug over one year while preserving drug potency. Jian et al. also developed a novel DDS, which combines natural and synthetic polymers to carry bevacizumab in microparticles [140]. They developed chitosan-polycaprolactone core-shell microparticles and tested the novel DDS in vitro and on ex vivo porcine eye models. The designed core-shell microparticles were able to increase the loading capacity by 25% and decrease the initial burst release to nearly 30%. The drug was released in vitro for 6 months and maintained drug potency. The nanotherapeutic proved to be biocompatible with over 90% cell viability. In vivo studies remain needed to assess safety and drug efficacy.

Gene delivery therapy was studied yet also remains below the 1-month drug release threshold. Chaharband et al. used gene therapy to deliver intravitreally VEGFR-2 siRNA in rabbit and rat laser model of CNV through chitosan-hyaluronic acid nano-polyplexes [141]. In vitro, the DDS suppressed VEGFR-2 expression by 70%, and it significantly reduced CNV in vivo after 14 days. The DDS could become an efficient intravitreal gene delivery therapy.

#### 6.1.3. Lipid-Based

Lipid-based nanocarriers have shown potential for co-loading multiple drugs, as demonstrated by Formica et al., who developed a hybrid lipid-based nanocapsule containing both bevacizumab on the surface and triamcinolone acetonide in the core. This co-loading approach offers the potential for more effective treatment of diseases, as seen with the reduction of both inflammation and neovascularization in the case of bevacizumab and TA. In vitro studies showed that this novel formulation significantly inhibited capillary formation, making it a promising drug delivery system for loading multiple drugs [142].

Liposomes are a promising drug delivery system (DDS) due to their hydrophilic core and hydrophobic outer shell, which can be modified to improve tissue penetration. However, the interactions of liposomes with macrophages, changes in pH, and enzymes can affect their performance, making it difficult to accurately predict their physiological behavior. In the study of Mu et al., they used multivesicular liposomes (MVLs) to encapsule bevacizumab [143]. The MVLs had a size ranging from 1 to 100 μm, which enabled them to not be captured by macrophages and, thus, be rapidly degraded. Liposomes are known for minimal toxicity, good biocompatibility, encapsulation efficacy, and low immunotoxicity, which was exhibited in this study. Bevacizumab was released by diffusion and erosion and kept its integral structure in vitro. After intravitreal injection in the laser-induced CNV rat model, the nanocarrier sustainably released the drug as opposed to the bevacizumab solution. After 28 days of treatment, the DDS could inhibit CNV lesions, unlike the bevacizumab solution. With these promising findings, this DDS could potentially reduce the frequency of intravitreal injections.

Kayland Karumanchi et al. similarly encapsulated bevacizumab in liposomes [144], but they obtained even longer prolonged drug release results. In vivo, the DDS maintained drug release at therapeutic levels for 22 weeks compared to less than 6 weeks for bevacizumab solution. Drug potency also remained preserved. Therefore, liposomes could offer a promising prolonged and controlled drug release.

Lastly, it was shown that liposomes could also become potential nanocarriers in cancer treatments. De Cristo Soares Alves et al. developed a chitosan-coated lipid core nanocapsules that transports bevacizumab to treat solid tumors like glioblastoma [145]. Within 24 h, bevacizumab and the DDS were compared in their ability to induce apoptosis. Bevacizumab alone did not significantly induce more apoptosis. Impressively, in chicken embryo chorioallantoic membrane (CAM), the drug-loaded nanocarrier used 5.6 times less doses of bevacizumab than in bevacizumab solution. The nanocarrier, thus, exhibits higher potent antiangiogenic effects. This shows that fewer drugs could be used in clinical practice, thus reducing high drug dose toxicity and adverse effects. With these promising results, this DDS has the potential to be used in the treatment of solid ocular tumors.

**Table 2 pharmaceutics-15-01094-t002:** Biodegradable nano-based DDS for anti-VEGF agents.

Disease	Drug	DDS	Advantages & Considerations	Administration Route	Stage	Reference
HYDROGEL						
CNV (AMD)	Bevacizumab	Thermo-sensitive hydrogel (mPEG-PLGA-BOX)	Prolonged drug release Easy preparation Good cytocompatibility and biodegradability Sol-gel phase transition No reported adverse effect on the retina	IVT injection	Preclinical: in vitro, in vivo (rabbit model)	[115]
CNV	Sunitinib Acriflavine	Liposome-loaded, injectable hydrogel (cSA@Lip-HAC)	Higher prolonged drug concentration Higher drug residency time Antiangiogenic activity Enhanced anti-CNV activity Anti-CNV activity superior to commerical product injection	Periocular (Subtenon)	Preclinical: in vitro, in vivo CNV rat model)	[120]
Wet-AMD, PDR	Aflibercept	PGLA microspheres in a PEG-PLLA-DA/NIPAAm thermorepsonsive hydrogel	Controlled and prolonged drug release over 6 months No reported toxicity Drug bioactivity remained at therapeutic level	Intravitreal injection	Preclinical: in vitro	[117]
CNV (AMD)	Conbercept	Short chain peptide hydrogel	Good biocompatibility Inhibits tube formation in human retinal endothelial cells Self-assemble as a hydrogel (pH-sensitive) Ease of injection	IVT injection	Preclinical: in vitro	[119]
CNV	Aflibercept	PGLA microspheres in a PEG-PLLA-DA/NIPAAm thermoresponsive hydrogel	Good tolerability and biocompatibility Prolonged drug release of 6 months No reported toxicity or ocular complications	IVT injection	Preclinical: in vivo laser-induced CNV rat model	[118]
CNV	Ranibizumab	PGLA microspheres in a PEG-PLLA-DA/NIPAAm thermorespon-sive hydrogel	Prolonged drug release over 6 monts Ease of injection Good biodegradability	-	Preclinical: in vitro	[38]
Wet-AMD, PDR	Bevacizumab Aflibercept	Thermoresponsive hydrogel PEG-PPG-PCL (EPC)	No reported toxicity to endothelial cells Good biocompatibility Controlled and moduable drug release rate of 40 days	IVT injection	Preclinical: in vitro, ex vivo rat model, in vivo rabbit model	[116]
CNV	Ranibizumab Aflibercept	PLGA microspheres in a PEG-PLLA-DA/NIPAA hydrogel PNIPAAm-PED-DA	Controlled and prolonged durg release No reported adverse effect Less drug required to achieve therapeutic effects Biocompatible Only studies the preventive effects of CNV	IVT injection	Preclinical: in vivo laser-induced CNV rat model	[114]
CNV (AMD)	Ranibizumab Aflibercept	PGLA in a PNIPAAm-PEG-diacrylate-thermoresponsive based hydrogel	Controlled and prolonged drug release for 196 days in vitro Bioactivity remained	IVT injection	Preclinical: in vitro	[113]
POLYMER: SYNTHETIC						
CNV (AMD), RNV (PDR, RoP)	Bevacizumab	PLGA nanoparticles	Higher drug bioavailability Prolonged drug release and drug half-life in vitreous and asqueous humor Increased antiangiogenic effects (HUVEC + OIR mice and CNV models) Decrease bevacizumab toxicity No reported toxicity on endothelial and retinalcells	IVT Subconjunctival	Preclinical: in vitro, in vivo alkali-induced CNV/OIR mouse model	[124]
CNV (AMD)	Bevacizumab Dexamethason	PLGA nanoparticles PVA Polyethilenimine	Stable in vivo Significantly decreased CNV leakage area Increased efficacy of CNV inhibition Antiangiogenic effects (in vitro HUVECs)	IVT	Preclinical: in vitro, in vivo	[127]
Wet-AMD	Bevacizumab	PDA nanoparticles	Antiangiogeneic activity of the DDS Reduces reactive oxygen species (ROS) associated with VEGF agents Controlled drug relrease	IVT	Preclinical: in vitro, ex vivo porcine eye model	[146]
Ocular angiogenesis	Bevacizumab-dextran	PLGA/PCADK microspheres	Prolonged drug release over 50 days in vitro and in vivo Bioactivity remained No reported toxicity Good tolerability	IVT	Preclinical: in vitro, in vivo rabbit model	[129]
CNV (AMD)	Anti-VEGF agents	PEG CCS (S-PEG-ICG-RGD-RBZ nanoparticles) polymers	Decrease tube formation (HUVEC) Promising biosafety High drug bioavailability Good cellular compatibility	Intravenous	Preclinical: in vitro, in vivo laser-induced CNV mouse model	[132]
Wet-AMD	Bevacizumab	PLGA microspheres	Accurate prediction of drug transport and concentration	IVT injection	Preclinical: computational rabbit model	[128]
Wet-AMD	Aflibercept	PLGA nanoparticles	Prolonged drug release High encapsulation efficiency	-	Preclinical: in vitro	[125]
CNV (AMD)	Ranibizumab	Magnetic nanoparticles loaded PEG-PLGA copolymer (iron oxide (Fe3O4)/PEGylated poly lactide-co-glycolide (PEG-PLGA)	Tube formation inhibition (HUVEC) Increased antiangiogenic properties with maxtrix	-	Preclinical: in vitro	[126]
Wet-AMD	Bevacizumab	PGLA nanoparticules	Controlled and prolonged drug release Low release rate Drug bioactivity remained pH dependant drug release	-	Preclinical: in vitro	[122,123]
CNV (AMD), PDR, RVO	Sunitinib	PLGA-PEG microparticles	Non-inflammatory Nontoxic Prolonged drug release over 6 months in vivo mouse model	Intravitreous injection	Preclinical: in vitro, in vivo laser induced CNV mouse models	[130]
Wet-AMD	Ranibizumab biosimilar	PLGA microparticles	Prolonged and controlled drug release Good target delivery	Intravitreous	Preclinical: in vitro	[121]
POLYMERS: SYNTHETIC AND NATURAL						
Wet-AMD	Bevacizumab	Chitosan PCL	Prolonged drug release and efficacy up to 6 months High loading capacity Drug potency remained up to 6 months Good biocompatibility Good injectability High physical integrity and uniform size	-	Preclinical: in vitro, ex vivo porcine eye model	[140]
Wet-AMD	Bevacizumab	PCL-chitosan bi-layered capsule	Good structural integrity Low toxicity Maintain drug efficacy Controlled and prolonged drug release over one year in vitro	IVT injection	Preclinical: in vitro	[139]
Wet-AMD, PDR	Bevacizumab	Chitosan PLGA nanoparticles	Controlled and prolonged drug release Increased surface and posterior drug bioavailability	Subconjunctival injection	Preclinical: in vitro, ex vivo pig/ goat	[136]
POLYMER: NATURAL						
CNV	Bevacizumab Suramin	Human serum albumin nanoparticles	Prolonged and controlled drug release Accommodate with hydrophilic drugs	Topical	Preclinical: in vitro	[135]
CNV	Bevacizumab	Human serum albumin nanoparticles	Prolonged drug release Stays on the eye surface 4 h post-topical application Increased antiangiogenic efficacy (rat CNV model) Formulation stability No reported toxicity	Topical	Preclinical: in vitro, in vivo CNV rat model	[133,134]
CNV	Bevacizumab	Chitosan nanoparticles	Excellent biocompatibility No toxicity reported Drug structure remains	-	Preclinical: in vitro	[55]
CRVO, CME, CNV (AMD), PDR)	Bevacizumab	Chitosan grafted poly(ethylene glycol) methacrylate (PEGMA) nanoparticles	No reported toxicity Good hemocompatibility Ocular tolerability Conrolled and prolonged drug release between 14–30 days Lower drug quantitiy while preserving therapeutic level	Subtenon	Preclinical: in vitro, in vivo rabbit model	[138]
Ocular NV	Bevacizumab	Chitosan nanoparticles	Controlled and effective drug concentration High drug concetnration Maintain drug efficacy No reported toxicity adverse effect Unequal drug loading capacity Unknown stability of nanoparticles	Subtenon	Preclinical: in vitro, in vivo rabbit model	[137]
CNV (AMD)	VEGFR-2 siRNA	Chitosan-hyaluronic acid nano-polyplexes	No reported toxicity Prolonged drug release Low water solubility may hinder DDS stability and loading capacity	Intraviteral injection	Preclinical: in vivo laser-induced CNV rat eye and rabbit models	[141]
LIPID-BASED						
Solid tumors (GB)	Bevacizumab	Chitosan-coated lipid-core nanocapsules Gold	Potential cytotoxicity activity (in vitro) Potential kinetic instability Increased antiangiogenic effect	-	Preclinical: in vitro, in vivo	[145]
Ocular angiogene-sis	Bevacizumab Triamcinolone acetonide	Hybrid lipid nanocapsules	Increased antiangiogenic effects (HUVEC) High drug load capacity and antibody coupling capacity	-	Preclinical: in vitro	[142]
CNV (AMD)	Bevacizumab	Multivesicular liposomes (MLV)	Drug structure and efficacy remain Sustained drug release High encapsulation efficiency Bioactivity remained preserved Drug structural integrity preserved after release in vitro Potentical increased drug residency time in vitreous humor	IVT injection	Preclinical: in vitro, in vivo laser-induced CNV rat eye model	[143]
CNV (AMD), RNV (PDR)	Bevacizumab	Liposomes	Prolonged and slow drug release for 22 weeks in vivo Increased drug availability (rabbit model) Drug bioactivity remained Drug efficacy remained Increased antiangiogenic effect (CNV rat model)	IVT injection	Preclinical: in vitro, in vivo rabbit models	[144]

NP: nanoparticles. CNV: choroidal neovascularization. AMD: age-related macular degeneration. PLGA: Poly(lactide-co-glycolide). IVT: intravitreal. RNV: retinal neovascularization RoP: retinopathy of prematurity. HUVEC: human umbilical vein endothelial cells. OIR: oxygen-induced retinopathy. RVO: retinal vein occlusion. CME: cystoid macular edema. PEGMA: poly(ethylene glycol) methacrylate. MLV: multivesicular liposomes.DDS: drug delivery systems. mPEG-PLGA-BOX: methoxy poly(ethylene glycol)-poly(lactic-co-glycolic acid)-bioxirane. LNC: lipid-core nanocapsules. PDR: proliferative diabetic retinopathy. PGLA: poly(glutamic acid). PEG-PLLA-DA/NIPAA: poly(ethylene glycol)-poly(L-lactic acid)-diacrylate/N-isopropylacrylamide. PEG-PLLA-DA/NIPAAm: poly(ethylene glycol)-poly(L-lactic acid)-diacrylate/N-isopropylacrylamide. PEG-PPG-PCL: poly(ethylene glycol)-poly(propylene glycol)-poly(caprolactone). EPC: ethylene-propylene copolymer. PNIPAAm-PED-DA: poly(N-isopropylacrylamide)-poly(ethylene glycol)-diacrylate. PVA: polyvinyl alcohol. ROS: reactive oxygen species. VEGF: vascular endothelial growth factor. PDA: polydopamine. PCADK: poly(citric acid-dodecylamine-ketoglutaric acid). PEG CCS: poly(ethylene glycol) chitosan-coated silica. PCL: polycaprolactone. CRVO: central retinal vein occlusion. NV: neovascularization. VEGFR-2 siRNA: vascular endothelial growth factor receptor-2 small interfering RNA. GB: glioblastoma.

## 7. Pediatric Posterior Segment Diseases

Although the pediatric eye undergoes distinct physiological development, it is still comparable in terms of anatomy, and it is functional to the adult eye. In fact, most adult therapies can be applied to infants and children after adjusting the dosage [147]. Pediatric patients often require fewer drug doses to achieve similar therapeutic effects [147]. Prioritizing minimally invasive treatments in this population group is recommended to increase patient compliance. While topical administration routes are preferred, drug bioavailability significantly decreases due to anterior segment components acting as barriers. Therefore, nano-based drug delivery systems (DDS) have become an interesting avenue to increase drug bioavailability and prolong drug release.

However, despite similar eye anatomy and functions, important considerations remain in the conception and use of nanocarriers for pediatric patients [148]. Due to normal developmental growth, the pediatric eye is constantly interacting with other systems and, thus, undergoing physiological and anatomical changes [149]. Therefore, the pathophysiology of certain diseases compared to adults can vary. Moreover, infants and children have a higher tear film stability, whereby tear components differ between pediatric and adult populations in terms of proteins profiles and lipid conformations [150]. The amount of lacrimal content varies with age, with tear secretion believed to peak in youth before decreasing in adulthood. According to a systematic review and meta-analysis, children secrete moderately more tears than adults [151]. All these different factors need to be taken into consideration when nano-based DDSs are used towards pediatric patients. Table 3 presents a summary of recent relevant studies.

### Retinopathy of Prematurity

Retinopathy of prematurity (ROP) is an eye disorder that affects premature infants. It is caused by the disruption of the normal growth of blood vessels in the retina, leading to abnormal vessel growth on the surface of the retina. This condition can result in retinal detachment and blindness if left untreated, although most cases resolve without treatment. The treatment of Retinopathy of Prematurity (ROP) depends on the severity (i.e., Stages 1 to 5) and location (i.e., Zones 1 to 3) of the affected retina, as well as the presence of “plus disease.” If indicated, treatment should be administered within 72 h of diagnosis. The two most common treatments for ROP are laser therapy and anti-VEGF injections [152].

Laser therapy, while effective in reducing unfavorable outcomes, often requires general anesthesia or sedation, and 9–15% of patients still experience unfavorable outcomes post-treatment. Anti-VEGF injections, specifically bevacizumab, have shown to be more effective than laser therapy, particularly for the Zone 1 disease close to the optic disc and macula. However, anti-VEGF injections carry the risk of complications such as endophthalmitis, and damage to the retina or lens [71].

#### Current Landscape of Novel Nano-Based DDS

Gold nanoparticles (GNPs) succinate interest as they are known for their antiangiogenic properties [153]. Therefore, it drove researchers to use them in anti-VEGF drug delivery systems (DDS) [154]. Hyoung Kim et al. first studied the use of GNPs in vitro and in an oxygen-induced retinopathy (OIR) mouse model [155]. No drug was loaded or added within the nanocarrier. In a mouse model, retinal neovascularization was significantly inhibited, and no retinal toxicity was reported. It was, thus, shown that GNPs could supress the VEGFR-2 signaling pathway to inhibit retinal vascularization. However, GNPs are not normally easily biodegradable compared to other types of nanocarriers. Their clearance mostly relies upon cell cycle renewal, which can take time varying from weeks to months and, meanwhile, disturb physiological processes [156,157]. Despite poor biodegradability, new recent formulations could potentially improve it [158].

On the other hand, other types of nanocarriers have excellent biodegradability compared to GNPs. Lipid-based nanocarriers are known for their high bioavailability and simple formulation to industrialize, as well as their flexible loading capacity [159]. More recently, Bohley et al. showed promising results of a newly developed lipid-based nanocarrier that may prevent RoP [160]. Bohley et al. injected intravenously Cyclosporin A (CsA) encapsulated within lipid nanocapsules (LNC) in vitro and in mouse models. This study focused on developing RoP treatment for Phase 2. They decided to opt for the intravenous administration route given its limited adverse effects compared to intravitreal injections. They used LNC comparable to very low-density lipoproteins (VLDLs). These VLDLs extravagate the choriocapillaris and Bruch’s membrane to deliver lipids to the retinal pigment epithelium (RPE) [161], and the nanocarrier follows this same pathway mechanism. The LNC was composed of triglycerides, while the outer layer was made of phospholipids to allow the nanocarrier to permeate through the choriocapillaris cells. Nanocapsules were coated with a specific RPE-ligand, cyclo(-Arg-Gly-Asp-D-Phe-Cys) (cRGD), to increase target specificity. In vitro, it was demonstrated that both the cRGD ligand and phospholipids significantly increased target delivery specificity to the RPE.

As a result, one hour after the injection of the novel nanocarrier in healthy mice models, 2% of the total drug-loaded nanocarrier dose had accumulated in the retina. After a 5-day period, a significant amount remained, which indicated the formation of a depot. Additionally, CsA within the cRGD-LNC decreased neovascularization in OIR mice models by a factor of 6 by lowering VEGF and VEGF-R2 levels. Unlike anti-VEGF agents that completely inhibit VEGF levels, CsA normalizes VEGF to healthy markdowns, which is necessary to ensure retinal and posterior cell functioning. Therefore, CsA may avoid serious side effects that are encountered when completely inhibiting VEGF.

Additionally, anti-inflammatory properties were reported with significantly lower inflammation biomarkers following CsA–cRGD–LNC injections in OIR mice. While no adverse effect was reported on healthy mice, the nanocarriers were distributed throughout the entire body, which is a recurring issue with nanocarriers [162]. Despite this finding, the nanocarriers remain specific to RPE cells. Surprisingly, a single dose of CsA–cRGD–LNC injected directly after inducing hypoxia prevented the mice from developing RoP. Therefore, this nanotherapeutic could become a public health preventive treatment. Appropriate dosage and adverse effects still need to be explored.

Other studies have investigated gene delivery therapy through nano-based DDSs. Hagigit et al. injected the antisense oligonucleotide 17 (ODN17) inside an antisense oligonucleotide (DOTAP) cationic nanoemulsion to treat RoP in mice [163]. Antisense oligonucleotides (ODNs) can interfere with genes and, thus, in the case of RoP, potentially suppress VEGF expression. However, ODNs in vivo require protection from a nanocarrier to avoid rapid degradation in organs and by exonucleases. It was previously found that cationic nanoemulsion combined with DOTAP protects ODNs from degradation in vitro and in vivo [164,165]. When the DOTAP nanoemulsion was injected in RoP mouse models, an inflammatory reaction was reported. On the other hand, when the DOTAP nanoemulsion was combined with the ODN17, a minimal inflammatory reaction occurred. The efficacy of ODN17 was increased, which suggests two hypotheses: (1) The nanocarrier formulation stabilized ODN17 despite a high environmental nucleasic activity; and (2) The nanocarrier formulation increases ODN17 intracellular uptake.

This nanotherapeutic was tested topically and subconjunctivally on the corneal neovascularization-induced rat model. The subconjunctival injection inhibited 89% of the corneal neovascularization, while the topical administration route inhibited 83% of the corneal neovascularization. When tested on a RoP mouse model, the DDS inhibited 64% of the vitreal neovascularization. However, both administrative routes showed significant vascularization. Therefore, this DDS could also become an effective preventive and therapeutic approach to treat ocular neovascularizations.

Similarly, Xu et al. also explored gene delivery, but with the novel agent miR–24–3p expressed in the microglia-derived exosome to treat retinal photoreceptors [166]. They chose exosomes from microglial cells due to their involvement in retinal vascular development and potential antiangiogenic effect [167,168,169]. This was confirmed as microglia-derived exosomes decreased angiogenic factors, including VEGF and improved visual injury. MiR–24–3p are the therapeutic agents that prevent retinal photoreceptor apoptosis. Despite promising studies, an eminent challenge for this potential therapy is the difficulty to isolate primary human microglia-derived exosomes. However, upcoming cell extraction methods could facilitate this process [170]

Gene delivery therapy was further investigated by Wang et al., who used lipid-like nanoparticles to deliver siRNA in retinal cell cytosol to induce VEGF gene-silencing [171]. This novel therapeutic significantly decreased VEGF and protein expressions in vitro on HUVECs model and in vivo on a OIR rat model. When compared with standard bevacizumab injections, there were no reported differences between either treatment. While no adverse effects were reported in this study, previous clinical trials using RNA interference therapy were terminated due to major adverse effects [172]. Unlike this siRNA, previous studies had explored the “naked” siRNA bevasiranib. “Naked” siRNA is now known to prone nuclease degradation followed by activation of the immune system [173]. Thus, the use of nanoparticles is crucial to encapsulate the siRNA and protect it from degradation before its release in the targeted cells. The novel siVEGF lipid-like nanoparticles could become an efficient gene delivery DDS in retinal neovascularization diseases.

The most recent gene delivery study on RoP was conducted by Huang et al. They inserted miRNA-223 inside folic acid–chitosan-modified mesoporous silica nanoparticles (FA–CS–PMSN) in vitro and in oxygen-induced retinopathy (OIR) rat models [174]. This DDS increased the bioavailability and efficacy of miRNA-223, which normally is highly unstable and unspecific to target. Multiple components were added.

**Table 3 pharmaceutics-15-01094-t003:** Biodegradable nano-based DDS for retinopathy of prematurity.

Disease	Drug	DDS	Advantages & Considerations	Administration Route	Stage	Reference
ROP	Cyclosporin A (CsA)	Lipid nanocapsule (LNC)	Target cell specificity, but not exclusively (nanocarrier spreads in the entire body) High drug concentration Higher drug residency time No reported adverse effect Promising preventive treatment	Intravenous	Preclinical: in vitro, in vivo OIR mouse model	[160]
	miRNA-223	Folic acid–chitosan-modified mesoporous silica NPs	High stability Loading efficiency Targeted delivery Controlled drug release Reduces inflammation by transitioning to M2 phenotype	IVT injection	Preclinical: in vitro, in vivo mouse OIR model	[174]
	VEGF siRNA	Lipidoid NPs (1-O16B)	Inhibition of tube formation (HUVEC) In vivo results similar to ranibizumab therapy Lack of data for long-term outcomes Lack of data on side effects and immune reactions	IVT injection	Preclinical: in vitro, in vivo rat OIR model	[155]
	VEGF-A_165_	PLGA	Potential RoP Phase I treatment to avoid systemic effects of drugs targetting Phase II Controllable microparticle morphology Sustained drug release Reduction of retinal vaso-obliteration Prompt recovery of vein dilatation and arterial tortuosity Moderate protein loading capacity No reported systemic effects	IVT injection	Preclinical: in vitro, in vivo mouse OIR model	[175]
	miR-24-3p	Microglia-derived exosome	Low toxicity Inhibition of VEGF (in vitro and in vivo) Difficulty to identify and isolate human microglial cells	IVT injection	Preclinical: in vitro, in vivo mouse OIR model	[176]
	Anti-VEGF oligonucleotides (VEGFR-2 ODN17)	DOTAP blank cationic nanoemulsion	Potential drug stabilization with the nanoemulsion Inflammation reduction Enhanced efficacy in inhibiting neovascularization	IVT injection	Preclinical: in vivo ROP mouse and rat CNV models	[163,164,165]

Here, the nanoparticles gain new pharmacological properties. For instance, by adding a polyethylene glycol (PEG) chain to the miRNA-223, the nanocarrier was able to diffuse through the vitreous body. Folic acid enabled the nanocarrier to specifically target the microglial cells’ receptors. Chitosan polymers increased microglial cell permeability to avoid degradation by lysosomes. Through these newly acquired pharmacological characteristics, the nanocarrier was able to deliver bevacizumab and effectively reduce VEGF expression in vitro. In vivo on the OIR rat model, the standard bevacizumab solution inhibited 26.1% of the retinal neovascularization, while the novel DDS inhibited retinal neovascularization by 52.6%. MiRNA-223 also reduced inflammation by favoring the M2 anti-inflammatory phenotype in microglial cells both in vitro and in vivo. This shows that the DDS could also help in controlling and reducing inflammation. This novel gene delivery platform could thus become a therapeutic option in inflammatory-neovascularization diseases. Other experiments could be conducted to generalize anti-angiogenesis efficacy in other CNV diseases such as AMD.

So far, all the novel DDSs focused on treating RoP Phase II, which is characterized by hypoxia-induced neovascularization. In this study, Mezu–Ndubuisi et al. designed a poly(lactic-co-glycolic acid) (PLGA) nanocarrier to deliver the proangiogenic VEGF-A_165_ to the retina of RoP mouse models in Phase I [177]. RoP Phase I is characterized by lowered VEGF levels due to insufficient vascularization. Therefore, they hypothesized that by increasing VEGF levels in Phase I, Phase II could be prevented. PLGA microparticles have already proven to be efficient in transporting VEGF-A_165_ in vivo, which explains why it was chosen in this study. Findings showed that VEGF-A_165_-loaded PLGA microparticles reduced vaso-obliteration normally observed in Phase I and, thus, allowed retinal revascularization and accelerated vascular recovery phenotype. Therefore, this novel VEGF-A_165_-loaded PLGA microparticles nanotherapeutic could potentially become a therapy that escapes systemic adverse effects of phase II anti-VEGF drugs.

While promising nanotherapeutics could alleviate the physical and financial burdensome of this disease, some authors mention the lack of commercial interest to develop them [160]. In fact, there is a limited number of studies published on RoP compared to other adult ocular diseases. This could be explained by low RoP prevalence in developed countries and the existence of multiple RoP treatments already offered on the market [178]. Nevertheless, nanotherapeutics in RoP would revolutionize infant treatments by limiting anti-VEGF intravitreal injections, its associated complications and potentially could even prevent the development of this disease. Successful formula and drug trends include respectively lipid-based DDSs as they can easily penetrate tissue membranes and miRNA, which are being studied for their retinal neovascularization inhibiting properties [160,179]

Although the research on DDS in pediatric ophthalmology is still in preclinical phases, the field of nanotechnology in this area is relatively underdeveloped, as most of the research currently focuses on more prevalent adult ocular diseases [148]. Current nanocarriers are, thus, initially conceived and designed for adults. Moving forward, tailored clinical studies for pediatric patients will be required to find optimal drug dosage and ensure safety and efficacy.

## 8. Clinical Advancements in Posterior Segment Diseases

Current therapeutic trials exploring the use of biodegradable DDSs have focused on pre-clinical in vivo animal or in vitro cell models due to the difficulty in translating this research to humans. Despite these challenges, there have been several clinical studies evaluating the effectiveness of biodegradable drug delivery systems in treating posterior segment diseases. In this section, we will summarize the key findings and insights gained from these studies. The next section will explore the barriers to clinical translation in the context of biodegradable DDS for posterior segment diseases.

Implants are typically nondegradable, and in the event of complications, such as migration to the anterior chamber leading to corneal edema and decompensation, their removal necessitates open surgical procedures. The implant removal procedure constitutes an invasive treatment with potential surgical complications. Moreover, nondegradable implants remain in the vitreous and, if they migrate to the visual axis, may cause visual symptoms. There are degradable implant formulations available, which break down through erosion and/or biodegradation into water-soluble by-products, offering a better alternative than non-degradable implants [180]. Ozurdex^®^ is currently the only FDA-approved biodegradable implant for posterior segment delivery, demonstrating up to 4 months of sustained release. It contains the active drug, dexamethasone (DEX), dispersed in a PLGA polymer matrix and is intravitreally injected. It has received approval for macular edema following retinal vein occlusion, diabetic macular edema, and non-infectious uveitis. Castro–Navarro et al. reported that the implant was effective for diabetic macular edema, even in refractory patients who had failed to respond to previous treatment. In a single-centre Phase III prospective open-label study, Mathew et al. further observed that Ozurdex^®^ treatment had rapid effects on the macula, which were sustained for up to 8 weeks followed by modest effects up to 32 weeks. Patients experienced significant improvement in visual function by 24 weeks. They further suggested that the optimal retreatment point would be 20 weeks, highlighting a significant benefit of this therapy over conventional options, insofar as the need for continuous treatments [181]. DEX is further being assessed in PLGA formulations, and Posurdex^®^, a biodegradable intravitreal implant, is currently undergoing Phase III clinical trials [182]. In preclinical studies, DEX concentrations were sustained for 42 days following implantation with detectable levels for up to 56 days in rabbit eyes. In a Phase II clinical trial in patients with macular edema, statistically significant therapeutic results were achieved, with a higher drug dose of 700 µg demonstrating superior improvement in visual acuity [182]. Similarly, TLC399 (ProDex) containing DEX sodium phosphate, is currently in Phase II clinical trials for the treatment of macular edema due to RVO [183]. Other biodegradable nano-based implants for posterior segment diseases are still in preclinical phases, including the delivery of DEX through silicone or PLGA [121].

While other hydrogel-based nanocarriers are still in experimental phases, a Phase I study by Wong et al. is being conducted on a hydrogel-based implant for neovascular AMD [184]. This developed implant (OXT–TKI) is bioresorbable. It delivers axitinib, which is a tyrosine kinase inhibitor. The implant is formulated through micronized drug crystals in dry, biodegradable PLGA fibers [185]. Following the IVT injection, the system swells due to the hydration of the PLGA matrix, resulting in a rod-shaped implant. No adverse effects were reported by Wong et al. [184]. In some cases, the implant was durable for 13 months. Overall, the implant remained in a stable position and resorbed sustainably. The clinical trial is still active with promising results.

PLGA-based systems have likely seen the most success in clinical settings, as injectable drug-loaded and surface-treated PLGA microparticles are currently undergoing clinical trials for the delivery of sunitinib maleate (GB-102) to the posterior segment for wet AMD. Sunitinib maleate is an FDA-approved anti-cancer drug and has shown promise in preclinical cases [184,185]. A Phase IIb, 12-month randomized controlled clinical trial is underway in patients with wet AMD (NCT03953079), with initial results promising durability for 6 months after a single IVT injection. Trials evaluating GB-102 in patients with macular edema and retinal vein occlusion are also ongoing.

Drug-polymer conjugates are another effective approach to prolong drug residence time in ocular delivery. The anti-VEGF drug pegaptanib sodium (Macugen^®^) is the first biodegradable nano-based therapy approved for wet AMD. It is a PEG conjugated aptamer, demonstrating increased residence time and prolonged half-life [186].

Another ocular nanomedicine, Visudyne^®^ is an intravenous administration formulation for classic subfoveal CNV due to wet AMD. The active drug, verteporfin, is loaded with biodegradable PLGA nanoparticles, used in photodynamic therapy to eliminate abnormal blood vessels [183,184,185,186,187,188]. Several clinical trials are also underway to assess usefulness of Visudyne^®^ as an adjuvant to other therapeutic agents, including anti-VEGF agents and corticosteroids. However, the biodegradability of such formulations is compromised due to the non-biodegradable natures of the secondary therapeutic agents [182].

## 9. Translation of DDS from Bench to Bedside: Preclinical and Clinical Considerations

### 9.1. Barrier to Clinical Translation: Anatomical Differences across Animals and Humans and Their Implications for DDS in Posterior Segment Disease Treatment

This section focuses on evaluating topical drug delivery for retinal diseases, highlighting key considerations in translating preclinical models to humans.

Preclinical studies have reported successful topical delivery of small molecules or proteins to the retina using rodent models. However, caution is advised when interpreting efficacy data, since the anatomical dissimilarities between the rodent and human eye can impact drug distribution and elimination, affecting the pharmacodynamic response. For example, in humans, the sclera is relatively thin, making it more permeable to certain drugs. This allows drugs to penetrate into the eye more easily and reach the target tissues. However, in rodents, the sclera is much thicker than in humans. This can make it more difficult for drugs to penetrate the sclera and reach the target tissues. Another example is the size of the eye. Rodents have smaller eyes compared to humans, resulting in a smaller volume of aqueous humor. This can lead to higher drug concentrations in the eye after topical application and shorter residence time for drugs. Although rodent models have advantages such as established pharmacological models and practicality, the differences between the two species should be considered [189].

Preclinical studies using rodent models have shown promise in topically administered compounds for treating retinal diseases, but clinical trials have demonstrated the importance of choosing appropriate animal models. For instance, despite showing efficacy in rodent models, compounds like TG100801 and pazopanib failed to demonstrate efficacy in clinical trials for treating AMD. Similarly, acrizanib, a VEGFR-2 inhibitor that showed positive results in rodent models, failed to demonstrate efficacy in a proof-of-concept study in patients with neovascular AMD [11,190,191,192].

Rabbits have more similar anatomical and physiological parameters to humans compared to rodents, including ocular features such as size, vitreal volume, and internal structure [193]. They also have predictable correlations in intravitreal pharmacokinetic parameters with humans and are relatively easy and economical to handle as a larger species model. An increasing number of rabbit models of ocular diseases, including AMD, have been established [193]. However, preclinical data from rabbit models should be interpreted with caution due to anatomical and physiological differences that can affect drug disposition. These differences include a lower blinking rate in rabbits, which can increase the residence time of topically administered drug formulations and affect bioavailability in intraocular tissues [194]. The proportionally larger anterior segment and more viscous vitreous humor in rabbits compared to humans may also impact the distribution of drugs that rely on a corneal route of diffusion. Finally, pharmacokinetic studies typically use animals with healthy eyes, which can result in the underestimation of drug clearance when extrapolated to diseased human eyes with a compromised blood–retinal barrier [195].

In short, preclinical pharmacokinetic and efficacy data generated in rodent models may not be translatable to humans due to anatomical and fluid dynamic differences. Instead, larger species such as rabbits, dogs, pigs, or monkeys should be used to test novel drug delivery approaches, which will increase the likelihood of successful clinical translation.

### 9.2. Implications of Artificially Induced Disease Models in Clinical Translation

While promising results have been obtained, the long-term effects of advanced DDSs remain unknown. For example, most of the studies used laser-induced CNV animal models, but these models have been known to heal naturally over time [143]. Most of the studies did not evaluate their DDS on animal models with underlying diseases causing neovascularization (e.g., wet-AMD, PDR). Hence, prior to conducting clinical trials, data from animal models with underlying conditions leading to retinal neovascularization (rather than the artificially induced neovascularization) will be crucial. Evaluating DDS on these animal models will provide a more accurate representation of the potential benefits and limitations of DDS in treating posterior segment diseases.

### 9.3. Advancing Biodegradable Drug Delivery Systems for Ocular Applications: Insights and Future Directions

The field of biodegradable DDS showed promising results in preclinical studies as discussed previously using natural biomaterials and nanotechnology [196]. Additionally, particularly in the past decade, DDSs using RNA gene therapies have opened the door to potential new ways of treating ocular diseases [197]. An upcoming milestone to achieve is to better understand and characterize the pharmacokinetics, general behavior, and potential toxicities of these novel DDSs on in vivo healthy and non-ill-induced animal models. As discussed above, some ill-induced animal models naturally tend to heal over time. Therefore, assessing DDS efficacy on non-ill-induced animal models is crucial [143]. Upcoming clinical studies and in vivo research will expand our knowledge on the current promising ocular DDS. The use of safe biomaterials, chemical reactions, nanocomposites, and green synthesis techniques become an important modern consideration to reduce the environmental burden. Research in the field aims to develop eco-friendly and low-cost techniques, which can also be easily industrialized. There are some pharmaceuticals benefits to rely upon “green chemistry,” such as a safer DDS profile, stability, and higher biocompatibility [198]. Another explored way to innovate is by using synthetic and natural biodegradable polymer DDSs that can be customized in the laboratory to gain tailored properties. The combination of synthetic and natural polymers represents a promising avenue to increase biocompatibility and overall effectiveness [199].

## 10. Conclusions

In conclusion, the field of biodegradable nano-based drug delivery systems (DDS) holds immense promise in addressing the challenges posed by anatomical barriers in ocular drug delivery and the treatment of posterior segment diseases. This review article has provided a comprehensive examination of various aspects of this rapidly evolving field, including an overview of biodegradable nano-based DDS, as well as their application in the treatment of adult and pediatric posterior segment diseases.

The anatomical barriers in the eye, such as the complex structures and robust static and dynamic barriers, limit the penetration, residence time, and bioavailability of topical and intraocular medications. Biodegradable nano-based DDSs can stay in ocular tissues for longer periods of time, increasing bioavailability and reducing the frequency of drug administration. Additionally, these systems can be made up of biodegradable polymers that are nanosized, reducing toxicity and adverse reactions.

As we look toward the future, several research trends are poised to play a pivotal role in advancing biodegradable nano-based DDSs. These may include the exploration of novel biodegradable materials with superior biocompatibility and tailored degradation profiles, as well as the development of targeted DDS that improve site-specific drug delivery, such as the use of exosomes. Exosomes, as naturally occurring extracellular vesicles, can be harnessed for targeted drug delivery due to their inherent ability to transport biomolecules and their potential to be engineered for cell-specific targeting. Furthermore, integrating stimuli-responsive elements into these systems could enable controlled release of therapeutics in response to specific physiological conditions or external triggers.

Before we can fully realize the potential of these research trends, there are several challenges to overcome, one of which is the translation of findings from animal and preclinical models to humans. This necessitates the development of reliable and predictive models that accurately reflect human ocular physiology and disease pathogenesis. A better understanding of the differences between animal and human models, as well as the development of advanced in vitro and in silico models, can help bridge this translational gap. Moreover, interdisciplinary collaboration between material scientists, pharmacologists, and ophthalmologists will be essential in the translation of these innovative systems from the laboratory to clinical practice. By working together and leveraging each field’s expertise, researchers can develop more effective and safer biodegradable nano-based DDS for ocular applications.

In summary, the advances in biodegradable materials combined with a better understanding of ocular pharmacology have allowed for the rapid evolution of biodegradable nano-based DDSs, offering great promise to overcome the current challenges encountered by an ophthalmologist in the treatment of posterior segment diseases. This review highlights the importance of biodegradability in the development of effective drug delivery systems for the eye and underscores the potential for further advancements in this field.

## Figures and Tables

**Figure 1 pharmaceutics-15-01094-f001:**
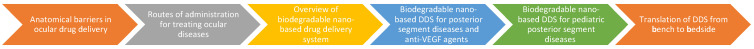
Flow chart graphical abstract. This figure provides a visual representation of the main points and structure of our article through a flow chart graphical abstract. It presents an overview of the topics that will be covered in the article and the sequence in which they will be presented.

**Figure 2 pharmaceutics-15-01094-f002:**
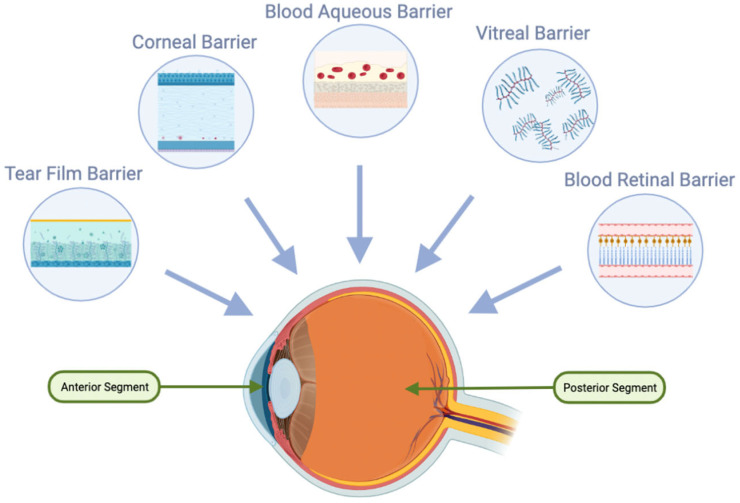
Anatomical barriers in ocular drug delivery. This figure illustrates the delivery barriers that drugs encounter when trying to reach the ocular tissues. These barriers include the tear film, corneal epithelium, blood-aqueous barrier, vitreous barrier, and blood-retinal barrier. Together, they significantly reduce drug availability in the ocular tissues of the posterior segment.

**Figure 3 pharmaceutics-15-01094-f003:**
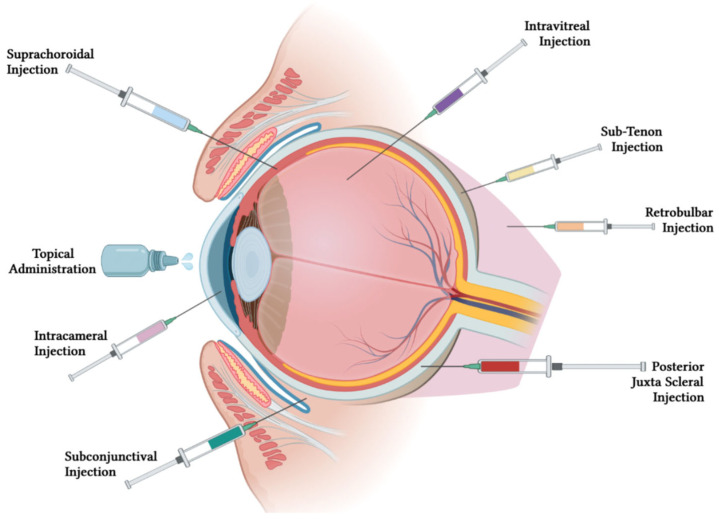
Ophthalmic medication delivery routes. This figure provides an overview of the various routes by which medications can be delivered to the eye. The figure shows the different routes of administration, including topical, subconjunctival, suprachoroidal, intracameral, intravitreal, retrobulbar, sub-tenon, posterior juxta scleral, subretinal, and systemic administration.

**Figure 4 pharmaceutics-15-01094-f004:**
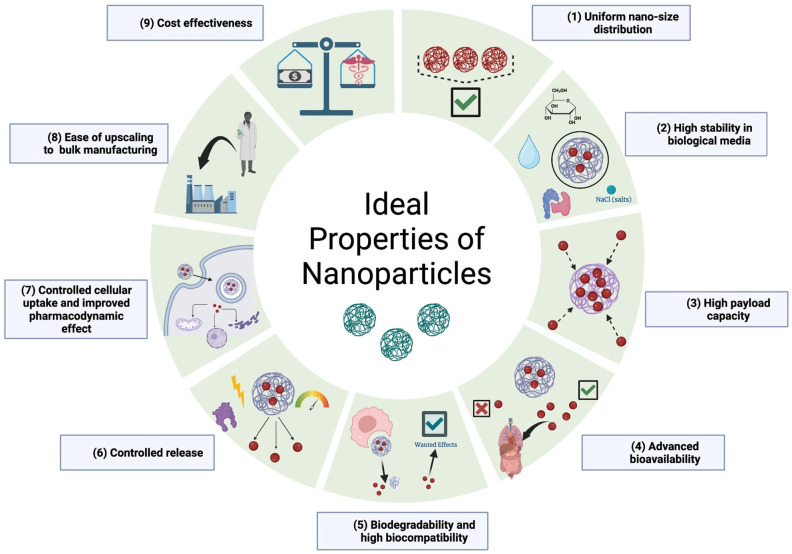
Ideal Properties of Nanocarriers. This figure illustrates the ideal characteristics of nanocarriers for drug delivery. Adapted by BioRender.com (2023). Retrieved from https://app.biorender.com/biorender-templates (accessed on 30 January 2023).

**Figure 5 pharmaceutics-15-01094-f005:**
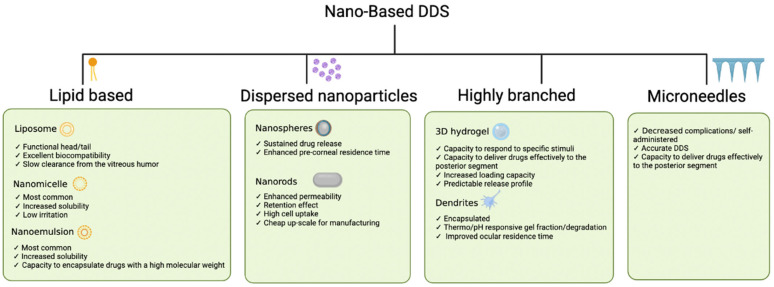
A comparative overview of characteristics and advantages of various drug delivery systems for ocular drug delivery. This figure provides a comparative analysis of different drug delivery systems used for ocular drug delivery. The figure illustrates the features and advantages of various biodegradable nano-based drug delivery systems. The figure also highlights the specific drug release mechanisms and advantages associated with each drug delivery system.

## Data Availability

Not applicable.

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
