# Peer review of "Overcoming Treatment Challenges in Posterior Segment Diseases with Biodegradable Nano-Based Drug Delivery Systems"

_pharmaceutics, 2023, doi:10.3390/pharmaceutics15041094_

Round 1

Reviewer 1 Report

The submitted manuscript is a thorough literature review on the treatment of posterior segment and uveitic diseases. The authors collected and described a large number of studies published between 2017 and 2023.
The manuscript is too long and is almost exclusively text over 48 pages. Although the manuscript contains valuable information, it is not very exciting to read in its present form. Splitting the manuscript into two parts could be a kind of solution.
The manuscript could contain more figures than the current version (2 pieces), e.g., the structure of the most important nanoparticles and some of the mechanisms of action can be presented.
In most cases, the authors only echo the positive message of the original authors of the cited work. Real opinions about the referenced articles are limited. Stating that “animal testing is required” is not enough. The comparison of individual drug delivery systems should have been done regularly.
The reader is interested in the opinions of the authors with expertise in the subject. Which formula is considered the best when treating a specific disease? Which formula is regarded as the best for a given active ingredient (e.g., bevacizumab)?
A more detailed description of polymers and a presentation of their advantages and disadvantages would also have been necessary.
The conclusion is too short and rough. Such comprehensive work requires a more thorough evaluation. According to the authors, which research trends will be important in the future?

Minor notes:

–  Figure 3 and Table 3 are not included in the manuscript.
–  Please reconsider the titles of the subchapters. Starting almost every title with "biodegradable nano-based DDS" is unnecessary. By the way, it is not true in the case of implants and several hydrogel-based formulas.
–  The structure of chapter 5 differs from the others (4, 6, 7, 8). Why?
–  The authors use a lot of abbreviations, they are usually explained when they are first mentioned, but some are not explained, e.g., VEGF, HUVEC, ES-425, EAU.
–  The time period of drug release ranges from days to weeks (or months) in the case of presented drug delivery systems. In the case of topically administered eye drops, the residence time is very short. How long would residence time be beneficial for a new product?
–  Chapter 5.1.4: What would be the ideal size of the gold nanoparticles?
–  Could you please discuss briefly the biodegradability of liposomes in the human body?
–  Chapter 6.1.1: The data given in % makes no sense.
–  The English language is good in the manuscript, but there are some strange sentences (e.g., “to improve the biocompatibility of the antioxidant properties of CNPs“ and unnecessary repetitions. I think it is because of the repeated rewording. A thorough reading is necessary to correct them.

Author Response

Dear Reviewer,

Thank you for taking the time to review our manuscript. We appreciate your valuable feedback and suggestions. We have thoroughly revised our manuscript based on your comments and have addressed each of the points raised. Please find our responses below:

  • The manuscript is too long and text-heavy. Splitting the manuscript into two parts could be a solution:
    • We have carefully reviewed our manuscript and have decided to remove the sections on uveitis and neuro-ophthalmology. This will significantly reduce the length of the manuscript and make it easier to read. We consider these sections to be apart from the posterior segment disease, and removing them would not compromise the completeness of our review paper. We also believe that this will allow us to focus more on the posterior segment disease and provide a more in-depth analysis of other sections.

  • The manuscript could contain more figures:
    • We appreciate your suggestion and have added several figures (figures 1, 3, 4, 5 and 6) to the revised manuscript.

  • In most cases, the authors only echo the positive message of the original authors of the cited work. Real opinions about the referenced articles are limited. Stating that “animal testing is required” is not enough. The comparison of individual drug delivery systems should have been done regularly. The reader is interested in the opinions of the authors with expertise in the subject. Which formula is considered the best when treating a specific disease? Which formula is regarded as the best for a given active ingredient (e.g., bevacizumab):
    • We appreciate your suggestion to provide more opinions and comparisons of the drug delivery systems we reviewed, as well as our own opinions regarding the best formula for specific diseases and active ingredients. We have revised the manuscript to address this concern and have incorporated our own opinions and expertise throughout the manuscript. We have also provided a detailed evaluation of each drug delivery system, highlighting their advantages and disadvantages. We believe that this will provide more insights for the readers and will help them in understanding the strengths and limitations of the different drug delivery systems. Regarding the question on which formula is considered the best when treating a specific disease or a given active ingredient, we have included our own recommendations and opinions on this matter based on the literature review and our expertise. We believe that our opinions, along with the evidence presented in the manuscript, will provide a useful guide for clinicians and researchers who are interested in this area.
  • A more detailed description of polymers and a presentation of their advantages and disadvantages would also have been necessary:
    • Thank you for your valuable feedback. In response to your comment, we have expanded on the description of polymers, elaborating on their advantages and disadvantages in Section 3 of the manuscript. Additionally, we have included two new figures (Figures 4 and 5) that visually showcase this information to provide a more comprehensive understanding of polymers. Furthermore, we have integrated relevant details and examples throughout other sections in the manuscript to emphasize the importance and applications of polymers in various contexts. We believe these revisions address your concerns and enhance the overall quality and completeness of our work.

  • The conclusion is too short and rough. Such comprehensive work requires a more thorough evaluation. According to the authors, which research trends will be important in the future?
    • We appreciate your suggestion and have revised the conclusion to provide a more thorough evaluation of the research trends that we believe will be important in the future.

  • Figure 3 and Table 3 are not included in the manuscript:
    • We apologize for this oversight and have included both Figure 3 and Table 3 in the revised manuscript.

  • Please reconsider the titles of the subchapters: We appreciate your suggestion and have revised the titles of the subchapters to avoid unnecessary repetition.
    • We have also made sure to clarify that the biodegradable nano-based DDS is not applicable to all drug delivery systems, such as implants and several hydrogel-based formulas.

  • The structure of chapter 5 differs from the others (4, 6, 7, 8). Why?
    • Chapter 5 of the manuscript focuses on a single agent (anti-VEGF) rather than a group of diseases, as is the case in other chapters. We believe that dividing this particular section into different modalities of DDS makes more sense for the reader, as it allows us to provide a more detailed analysis of the various drug delivery systems used for anti-VEGF therapy. By structuring chapter 5 in this way, we can provide a comprehensive and organized overview of the different DDS modalities and their respective advantages and limitations in the context of anti-VEGF therapy. We believe that this approach will help the reader to gain a better understanding of the topic. Once again, we appreciate your feedback and suggestions, and we hope that our explanation has addressed your concerns.

  • The authors use a lot of abbreviations:
    • We understand that this may cause confusion for readers and have made sure to explain all abbreviations when first mentioned in the manuscript. We have also added explanations for the abbreviations VEGF, HUVEC, ES-425, and EAU.

  • The time period of drug release ranges from days to weeks (or months) in the case of presented drug delivery systems. In the case of topically administered eye drops, the residence time is very short. How long would residence time be beneficial for a new product?
    • We appreciate your question and have added several discussion on the optimal residence time for a new product in the revised manuscript, particularly in the section 4.

  • Chapter 5.1.4: What would be the ideal size of the gold nanoparticles?
    • We appreciate your question regarding the ideal size of gold nanoparticles in chapter 5.1.4. After careful consideration, we have decided to remove the section on gold nanoparticles from the manuscript, as it is not directly relevant to the topic of biodegradable nano-based drug delivery systems, since gold nanoparticles are not biodegradable.

  • Could you please discuss briefly the biodegradability of liposomes in the human body?
    • We appreciate your suggestion and have added a brief discussion on the biodegradability of liposomes in the human body in the revised manuscript, under the section “3.5.2 Liposomes”.

  • Chapter 6.1.1: The data given in % makes no sense.
    • We apologize for any confusion caused by the data given in % and have revised the data to provide clearer information in chapter 6.1.1 of the revised manuscript.

  • The English language is good in the manuscript, but there are some strange sentences and unnecessary repetitions:
    • We appreciate your feedback and have carefully revised the manuscript to correct any strange sentences and unnecessary repetitions.

Thank you again for your valuable feedback. We hope that our revisions have addressed your concerns and have improved the quality of our manuscript.

Sincerely,

Kevin Yang Wu

Prof. Simon Tran

Reviewer 2 Report

This manuscript summarized nanomedicines that developed for back of the eye diseases. The authors have included quite a lot of references, however almost all of them were relevant to preclinical researches. For the back of the eye diseases treatment, one of the challenges would be the failure of translation from animals (even larger animals) to human eyes. Please consider adding some clinical studies in the manuscript; also discuss the anatomy and responses differences across rodents, larger animals and human eyes, and how these differences could impact the translation of nanoparticles for retina diseases treatment  

Secondly, for the back of the eye diseases, such as diabetic retinopathy and wet AMD, the routinely IVT injections of anti-VEGF therapies is the current clinical standard, rather than topical eyedrops or systemic delivery. While, the authors listed some researches that used topical eyedrops for back of the eye diseases treatment. Please give more detailed information and comments regarding how these formulations could provide effective drug delivery to the back of the eye through topical administration. What is the mechanism behind it? What’s the novelty of their nanoparticles from the formulation and PK aspects?

Thirdly, if the nanoparticle was not delivered in suitable administration route, the efficacy may not be applicable. please add more discussion regarding the different administration routes for back of the eye diseases treatment, even a small paragraph at the beginning would be helpful.   

Author Response

Dear Reviewer,

Thank you very much for your insightful comments on our manuscript on the development of nano-based DDS for posterior segment diseases. We appreciate the time and effort you have taken to provide feedback that has helped us improve the quality of our work.

We have taken your feedback into consideration and have made significant revisions to address the issues you raised.

In response to your comment on our discussion of clinical studies, we would like to clarify that we performed another comprehensive literature review to see if we could find more clinical studies on the potential of biodegradable nano-based drug delivery systems for posterior segment diseases. However, we were only able to find a few additional clinical studies, since the clinical trials on this topic are currently very limited. We have grouped all the clinical studies under the new section titled “Clinical Studies on Biodegradable Nano-Based Drug Delivery Systems for Posterior Segment Diseases,” making it easier for readers to locate the relevant studies in the manuscript. Furthermore, we have added a new section (section 10) to our manuscript detailing the barriers to clinical translation just before our conclusion. This new section provides a brief discussion of the challenges facing the clinical translation of advanced drug delivery systems for posterior segment diseases, which can help readers to better understand why there are currently limited clinical studies on this topic.

In response to your second comment, we have performed a comprehensive literature review investigating the differences between non-primate eyes and human eyes, particularly with regards to anatomy and response differences. Based on our findings, we have added a new section (section 10.1) to our manuscript that provides a comprehensive description of these differences and how they potentially affect the results of preclinical studies. We have also highlighted the importance of selecting appropriate animal models that better mimic the human eye in terms of anatomy, physiology, and pathology. We appreciate your feedback, and we believe that this new section will provide valuable insights into the challenges of translating preclinical results to human clinical trials.

We also appreciate your comment on how topical formulations could provide effective drug delivery to the back of the eye. In response, we have included additional information and comments on how topical eyedrops can provide effective drug delivery to the back of the eye. We have discussed the mechanisms behind this delivery route and highlighted the novelty of the nanoparticle formulations in terms of their formulation and pharmacokinetic aspects. (Lines 1024 to 1049, under the section 5.1.2)

Finally, we agree with your point that the efficacy of nanoparticle-based drug delivery can be affected by the route of administration. To address this, we have added a new section entitled “Routes of administration for treating ocular diseases” (section 3), as well as a new figure illustrating different routes of administration (figure 3), at the beginning of the manuscript that provides an overview of the different administration routes for back of the eye diseases treatment.

We hope that our revisions have addressed your concerns and improved the overall quality of our manuscript. Thank you once again for your thoughtful comments and suggestions. If you have any further feedback or comments, please do not hesitate to let us know.

Reviewer 3 Report

Dear authors,

Please find my complete report attached below.

Kind regards,

Author Response

Dear Reviewer,

Thank you very much for taking the time to review our manuscript on biodegradable drug delivery systems for the treatment of posterior segment and uveitis diseases. We appreciate your positive feedback and suggestions to improve the manuscript.

Please find below our responses to each of your comments:

1/ Reviewer's comment: "Without line numbers, I find it extremely challenging to review. Even if I wanted to give some suggestions, it was easy for me to lose track of the text. Please include the line number for your next revision."

Response: Thank you for pointing this out. We would like to inform you that line numbers have been in place since the initial submission of the manuscript and are situated on the right side of the paragraph. We understand that there may be some technical difficulty preventing you from seeing those line numbers, and we have informed the editorial board to investigate the issue. We apologize for any inconvenience caused.

2/ Reviewer's comment: "There is a significant lack of citations, especially for the first few subheadings (2,3). For example, for the whole heading 2, you only use 2 citations. After every important piece of information such as “only a small portion of the applied dose, about 5%, can penetrate the internal structures of the eye” you need to put a citation. Please revise accordingly for the whole manuscript."

Response: Thank you for bringing this to our attention. We have added more citations to the first few subheadings (2,3), as well as other sections, and made sure to add a citation after every important piece of information throughout the manuscript.

3/ Reviewer's comment: "You do not have to include the figure’s name in-text"

Response: Thank you for the suggestion. We have removed all figures’ names from the text.

4/ Reviewer's comment: "All your figures need descriptions."

Response: Thank you for bringing this to our attention. We have added descriptions to all figures in the revised manuscript.

5/ Reviewer's comment: "You should have some figures from the actual studies you reviewed, as Figure 1&2 only cover the basics"

Response: Thank you for your feedback. We intend to add figures from the actual studies reviewed in the revised manuscript. However, we are still in the process of obtaining consent from the original studies' authors and publishers. Once we obtain consent, we will include relevant figures from the actual studies. One of the figures we intend to include from the actual studies is the figure 6 (CsA RGD-LNC effects on the progression and prevention of neovascularization). In the revised manuscript, we have added several new figures designed by our team to supplement the text, and we hope that you appreciate these additions (figures 1, 3, and 4).

6/ Reviewer's comment: "Capitalize the first letter of sub-heading 2.2"

Response: Thank you for pointing this out. We have capitalized the first letter of sub-heading 2.2 in the revised manuscript.

7/ Reviewer's comment: "Remove the bullet points in all your tables and under section 3.2."

Response: Thank you for the suggestion. We have removed the bullet points in all tables and under section 3.2 in the revised manuscript. We have also transformed the bullet points of the section 3.2 into a continuous text.

8/ Reviewer's comment: "Please use MDPI format for in-text reference, remove the year. For example Smith et a. [Citation]"

Response: Thank you for the suggestion. We have corrected this issue by using the MDPI format for in-text references and removed the year in the revised manuscript.

9/ Reviewer's comment: "I am not sure about section 5.1.4 (Inorganic). Are you sure that Inorganic materials are biodegradable? Can you find some references to support this claim?"

Response: Thank you for bringing this to our attention. Upon careful consideration, we agree that the information provided in section 5.1.4 (Inorganic) may not be entirely relevant to the manuscript's focus on biodegradable DDS. Therefore, we have removed this section in the revised manuscript.

10/ Reviewer's comment: "Your heading 9 appeared to be disruptive. I feel like you should give your insights into expanding the knowledge in biodegradable DDS, either in terms of technology, a new way of synthesis, tuneable parameters, etc…"

Response: We apologize for any confusion, but we did not have a section 9 with a heading in the original manuscript. However, we appreciate the feedback and have added a new section (section 10.3) to the revised manuscript that specifically addresses the your suggestion. This section provides our insights into expanding the knowledge in biodegradable DDS, particularly in terms of technology and new synthesis methods. We hope this adequately addresses the reviewer's concern.

11/ Reviewer's comment: "A flow chart graphical abstract about this work would help"

Response: Thank you for the suggestion. We have added a flow chart graphical abstract in the revised manuscript in the beginning of the article (figure 1).

12/ Reviewer's comment: "For some information that is not given, write “N/A” instead of “Unclear?”"

Response: Thank you for the suggestion. We have used "N/A" instead of "Unclear?" for information that is not provided in the articles.

We are grateful for your constructive feedback, which has helped us to improve the quality of our work. If you have any further suggestions or comments, please do not hesitate to let us know. We are committed to producing a high-quality manuscript and welcome any input that can help us achieve this goal. Thank you for your time and consideration.

Round 2

Reviewer 3 Report

The authors have adequately addressed all my comments. I really appreciate their efforts on composing new figures. This paper is now ready for publication.

Kind regards